# Heroin Self-Administration and Extinction Increase Prelimbic Cortical Astrocyte–Synapse Proximity and Alter Dendritic Spine Morphometrics That Are Reversed by N-Acetylcysteine

**DOI:** 10.3390/cells12141812

**Published:** 2023-07-08

**Authors:** Benjamin M. Siemsen, Adam R. Denton, Jeffrey Parrila-Carrero, Kaylee N. Hooker, Eilish A. Carpenter, Meagan E. Prescot, Ashley G. Brock, Annaka M. Westphal, Mary-Nan Leath, John A. McFaddin, Thomas C. Jhou, Jacqueline F. McGinty, Michael D. Scofield

**Affiliations:** 1Department of Anesthesia and Perioperative Medicine, Medical University of South Carolina, Charleston, SC 29425, USA; 2Department of Neuroscience, Medical University of South Carolina, Charleston, SC 29425, USA

**Keywords:** heroin, astrocytes, GFAP, N-acetylcysteine, prelimbic, dendritic spines

## Abstract

Clinical and preclinical studies indicate that adaptations in corticostriatal neurotransmission significantly contribute to heroin relapse vulnerability. In animal models, heroin self-administration and extinction produce cellular adaptations in both neurons and astrocytes within the nucleus accumbens (NA) core that are required for cue-induced heroin seeking. Specifically, decreased glutamate clearance and reduced association of perisynaptic astrocytic processes with NAcore synapses allow glutamate release from prelimbic (PrL) cortical terminals to engage synaptic and structural plasticity in NAcore medium spiny neurons. Normalizing astrocyte glutamate homeostasis with drugs like the antioxidant N-acetylcysteine (NAC) prevents cue-induced heroin seeking. Surprisingly, little is known about heroin-induced alterations in astrocytes or pyramidal neurons projecting to the NAcore in the PrL cortex (PrL-NAcore). Here, we observe functional adaptations in the PrL cortical astrocyte following heroin self-administration (SA) and extinction as measured by the electrophysiologically evoked plasmalemmal glutamate transporter 1 (GLT-1)-dependent current. We likewise observed the increased complexity of the glial fibrillary acidic protein (GFAP) cytoskeletal arbor and increased association of the astrocytic plasma membrane with synaptic markers following heroin SA and extinction training in the PrL cortex. Repeated treatment with NAC during extinction reversed both the enhanced astrocytic complexity and synaptic association. In PrL-NAcore neurons, heroin SA and extinction decreased the apical tuft dendritic spine density and enlarged dendritic spine head diameter in male Sprague–Dawley rats. Repeated NAC treatment during extinction prevented decreases in spine density but not dendritic spine head expansion. Moreover, heroin SA and extinction increased the co-registry of the GluA1 subunit of AMPA receptors in both the dendrite shaft and spine heads of PrL-NAcore neurons. Interestingly, the accumulation of GluA1 immunoreactivity in spine heads was further potentiated by NAC treatment during extinction. Finally, we show that the NAC treatment and elimination of thrombospondin 2 (TSP-2) block cue-induced heroin relapse. Taken together, our data reveal circuit-level adaptations in cortical dendritic spine morphology potentially linked to heroin-induced alterations in astrocyte complexity and association at the synapses. Additionally, these data demonstrate that NAC reverses PrL cortical heroin SA-and-extinction-induced adaptations in both astrocytes and corticostriatal neurons.

## 1. Introduction

Relapse remains a major clinical obstacle in treating opioid use disorders and is typically brought on by a craving evoked by exposure to opioid-associated contexts or cues [1]. In humans, elevated activity in the corticostriatal circuit is a neural correlate of cue-evoked craving for heroin [2], and heroin-mediated dysfunction in corticostriatal circuity is faithfully recapitulated in rodent models of heroin self-administration (SA) and relapse [3]. A general consensus from rodent models is that the dysfunctional prefrontal cortical regulation of ventral striatal plasticity occurs as a result of chronic opioid exposure, allowing opioid-associated cues and contexts to drive drug seeking even after prolonged periods of abstinence [4].

The nucleus accumbens core (NAcore) is a ventral striatal region required for motivationally relevant external stimuli to evoke motor programs linked to seeking reinforcers or rewards [5]. It has been repeatedly demonstrated that heroin SA in rodents alters glutamatergic plasticity in the NAcore in a manner that is mechanistically linked to relapse [6,7]. For example, following the extinction of heroin [8] or cocaine [9] SA, an inability to induce long-term potentiation (LTP) and long-term depression (LTD) in the NAcore is observed following stimulation of the prelimbic (PrL) cortex in vivo. These data demonstrate drug-mediated dysfunction at PrL-NAcore synapses following withdrawal from addictive drugs that fundamentally alters synaptic plasticity. Akin to cocaine [10], heroin SA and extinction produce reduced glutamate clearance in the NAcore due to the downregulation of the high-affinity astrocytic glutamate transporter, GLT-1 [11], and concomitant reductions in the proximity of GLT-1-containing perisynaptic astrocytic processes (PAPs) to NAcore synapses [12]. Interestingly, decreased astrocytic association with NAcore synapses is also observed following the extinction of methamphetamine or cocaine SA [13,14]. Together, these data suggest that structural and cellular astrocyte dysfunction in the NAcore is a central aspect of the pathophysiology underlying relapse vulnerability [15]. As such, the reduced GLT-1 expression coupled with decreased proximity of astrocytic PAPs to synapses likely synergistically facilitate glutamate spillover to extrasynaptic sites following synaptic release. Consistent with dysfunctional plasticity at PrL-NAcore synapses, cue- and heroin-primed seeking promotes the sustained elevation of glutamate levels in the NAcore specifically due to the release from PrL cortical afferents [16]. This elevated glutamate release and spillover facilitates LTP-like increases in synaptic strength, including an upregulation of NR2B-containing NMDA receptors and enlargement of dendritic spine heads on NAcore medium spiny neurons (MSNs). Accordingly, it has been shown that normalizing drug-induced alterations in PrL-NAcore synaptic transmission, mediated in large part by astrocytic dysfunction, is a potent means of decreasing relapse vulnerability [17].

One drug that is particularly effective at targeting addictive drug-induced adaptations in astrocytes is the antioxidant, N-acetylcysteine (NAC). NAC is an L-cysteine pro-drug that has shown some promise clinically [18] but has been more extensively investigated preclinically as a research tool to probe glutamate homeostasis [15]. NAC has been shown to significantly limit heroin [19,20], cocaine [21,22], nicotine [23], or methamphetamine [14] seeking in rats with a drug self-administration history. Mechanistically, NAC is thought to prevent relapse by reversing the drug-induced downregulation of GLT-1 in the NAcore. This hypothesis has been confirmed in studies in which cocaine was used as a means to downregulate the expression of GLT-1 in the NAcore. In these studies, preventing the NAC-mediated restoration of GLT-1 expression in the NAcore following cocaine SA and extinction blocked NAC’s ability to prevent relapse [10,24]. While NAC does indeed facilitate extinction and prevent cued heroin seeking [19], surprisingly, the ability of NAC to normalize heroin-induced decreases in GLT-1 in the NAcore remains to be directly evaluated. However, additional drugs that upregulate GLT-1 expression (i.e., ceftriaxone) have been shown to normalize heroin-induced decreases in glutamate clearance in the NAcore, inhibiting relapse [11]. Importantly, ceftriaxone has also been shown to normalize cocaine-induced decreases in PAPs’ association with synapses in the NAcore [13], suggesting a significant structure–function relationship underlying drug-induced decreases in PAP–synapse associations, alterations in synaptic glutamate clearance, and relapse.

While there is a substantial body of work regarding the consequences of heroin SA and extinction on neuronal and astrocyte plasticity in the NAcore, little is known about the impact of heroin on neurons or astrocytes in the PrL cortex. This gap in our knowledge is particularly significant given that cue-induced glutamate release in the NAcore during heroin seeking requires THE activation of the PrL cortex [16]. While less is known about the relationship between cortical astrocytes and opiates, there is some literature describing opiate-induced adaptations in cortical astrocytes. For example, chronic morphine treatment increases the expression of glial fibrillary acidic protein (GFAP), a commonly used marker for astrocytes [25], in the frontal cortex [26]. These data suggest that morphine induces astrogliosis, yet the addition of morphine to astrocytes in vitro decreases the proliferation and increases the complexity of astrocyte processes, an effect prevented by naloxone pretreatment [27]. These data suggest that heroin may impact astrocytic function in the PrL cortex; however, such adaptations have yet to be shown in vivo or linked to the degree to which astrocytes are associated with synapses.

Here, we first employ electrophysiological recordings to interrogate functional adaptations in PrL astrocytes following heroin SA and extinction. Next, we employ immunohistochemical preparations in conjunction with confocal microscopy and 3D rendering to investigate the effects of heroin SA and extinction upon the GFAP cytoskeletal complexity of PrL astrocytes in addition to the PAP association with synaptic markers of NAcore projecting PrL neurons. We also investigate the therapeutic efficacy of NAC in reversing heroin SA-and-extinction-induced alterations in neuronal and astrocyte morphology. Finally, we investigate the effects of the inhibition of PrL thrombospondin 2 (TSP-2) in the context of NAC treatment following heroin SA, extinction, and cued-relapse to heroin seeking as a means to determine if NAC exerts its anti-relapse effects by engaging astrocyte-mediated synaptogenesis.

## 2. Materials and Methods

### 2.1. Animal Subjects and Surgery

Male Sprague–Dawley rats (Experiment 1: *N =* 22, Experiments 2–3: *N* = 63, Experiment 4: *N =* 20) were purchased from Charles River Laboratories (Wilmington, MA, USA) and were single-housed upon arrival in a temperature- and humidity-controlled vivarium with standard rat chow (Harlan; Indianapolis, IN, USA) and water available ad libitum. Rats were maintained on a 12 hr reverse light/dark cycle (lights off at 6 a.m.). All animal use protocols were approved by the Institutional Animal Care and Use Committee of the Medical University of South Carolina and were performed according to the National Institutes of Health Guide for the Care and Use of Laboratory Animals (8th ed., 2011). At the time of surgery, all rats weighed 275–300 g. On the day of surgery, rats were anesthetized with an intraperitoneal (i.p.) ketamine (66 mg/kg) and xylazine (1.33 mg/kg) injection, and received ketorolac (2.0 mg/kg, i.p.) for analgesia. A silastic catheter was implanted in the right jugular vein, attached to a cannula, which exited the animal’s back. Rats in Experiment 1 received a bilateral intra-PrL cortical infusion of AAV5.*GfaABC1D*::tdTomato (0.75 µL/hemisphere, titer: 7 × 10^12^ viral genomes per mL (vg/mL); co-ordinates- AP: +2.8, ML: +/−0.6, DV: −3.8 mm from bregma). Rats in Experiment 2 received a bilateral intra-PrL cortical microinjection of AAV5.*GfaABC1D*::Lck-EGFP (0.75 µL/hemisphere, titer: 1 × 10^13^ vg/mL) to label the fine membrane processes of astrocytes [13,28]; here, the truncated Lck was used, i.e., only the first 26 amino acids of Lck lacking kinase activity but allowing plasma membrane targeting of appended EGFP. Rats in Experiment 3 received a bilateral microinjection of AAV1.*CAG*.Flex::Ruby2sm-FLAG (Titer: 5 × 10^12^ vg/mL) as well as intra-NAcore (co-ordinates-AP: +1.7, ML: +/−1.6, DV: −7 mm relative to bregma) injection of rgAAV.*hSyn*::Cre-WPRE.hGH (0.75 µL/hemisphere, titer: 7 × 10^12^) to specifically target PrL cortical-NAcore neurons for dendritic spine analyses [29]. Rats in Experiment 4 received bilateral PrL injections of either AAV5 *GfaABC1D*::Lck-EGFP (titer: 1 × 10^13^ vg/mL) or AAV5. *GfaABC1D*::shTSP2-GFP. Injections (0.75 µL/hemisphere) were performed over a period of 5 min (0.15 µL/min) using a Nanoject II (Drummond Scientific, Broomall, PA, USA) and injectors were left in place for 5 min to facilitate viral diffusion from the injection site. Apart from the TSP2 RNAi construct, a gift from the Dong Lab [30], all viral constructs were purchased from Addgene (Cambridge, MA, USA). Additionally, the pAAV5. *GfaABC1D*::Lck-EGFP construct was purchased as a plasmid from Addgene, then packaged to produce viral particles at the University of North Carolina Gene Therapy Center viral vector core (Chapel Hill, NC, USA). Rats were allowed at least 5 days of post-operative recovery during which catheters were flushed daily with taurolidine citrate solution (TCS, Access Technologies, Skokie, IL, USA) and food and water were available ad libitum. For each experiment, ≥2 independent cohorts were used. Figure 1 illustrates the experimental design and timeline for all experiments. Appendix A contains names, descriptions, and sourcing information for all viral constructs used in the present experiments.

### 2.2. Drugs

Heroin hydrochloride was provided by the National Institute on Drug Abuse. N-acetylcysteine was purchased from Sigma-Aldrich (St. Louis, MO, USA, #A7250) and dissolved in sterile saline at 100 mg/mL.

### 2.3. Heroin Self-Administration, Extinction, and Cued-Reinstatement Procedures

All experiments were performed between the hours of 8 a.m. and 6 p.m. All SA experiments were performed in standard MedPC operant chambers equipped with two retractable levers, a tone generator, and a cue light (Med Associates, Fairfax, VT, USA). The beginning of each session was signaled by illumination of the house light and lever availability. Rats were trained for 14 days (3 h/day) to lever-press for heroin on a fixed ratio 1 (FR1) schedule of reinforcement, whereby each individual lever press on the active lever (ALP) resulted in the presentation of cue delivery and infusion of heroin followed by a 20 s timeout whereby ALP elicited no drug delivery. On days 1–2, active ALP resulted in a light and tone cue delivery followed by a 100 μg/50 μL i.v. infusion of heroin hydrochloride dissolved in sterile saline. On days 3–4, rats received a 50 μg/50 μL infusion followed by a 20 s. timeout period, and on days 5–14, rats received a 25 μg/50 μL infusion. Yoked-saline (YS) control animals were paired to a randomly selected heroin SA animal, receiving non-contingent saline infusions in a temporally paired manner. SA criteria were set at ≥1 infusion per session on days 1–4 whereas animals were required to receive ≥10 infusions per session on days 5–14. Following SA, all rats entered extinction whereby ALP resulted in no cue or drug delivery. All rats received 12–13 days of extinction training (3 h/day). Beginning on day 4, a subset of animals received N-acetylcysteine (100 mg/kg, i.p.) or vehicle (0.9% *w*/*v* NaCl) 30 min before each subsequent session, and all animals injected with N-acetylcysteine received 13 days of extinction training (Experiments 2–4). All YS animals received vehicle injections during extinction. This dose of NAC has previously been shown to reduce extinction responding on the active lever and provide an enduring reduction in cued and primed heroin seeking using the same heroin SA protocol [19]. Cued-reinstatement testing (Experiment 4) occurred following the final day of extinction training (3 h) in which ALP resulted in a light and tone cue delivery without the delivery of heroin.

### 2.4. Whole-Cell Recordings from Astrocytes

Twenty-four hours after the last extinction session, rats were anesthetized with isoflurane, decapitated, and their brains quickly removed and placed in an ice-cold and oxygenated artificial CSF (aCSF), slicing solution containing (in mM): 126 NaCl, 2.5 KCl, 1.2 MgCl_2_, 1.4 NaH_2_PO_4_, 25 NaHCO_3_, 11 D-glucose, and 0.4 ascorbate, supplemented with kynurenic acid (2.6 mM). Thick sections (∼220 μm) containing the PrL region were prepared using a vibrating microtome (Leica Biosystems, Deer Park, IL, USA) and maintained in oxygenated aCSF in a dish in a 32 °C water bath until recording. During recording, sections were perfused at a flow-rate of ~2 mL/min with oxygenated aCSF at 33 °C. The prelimbic layer V/VI was identified initially using a 4× objective (0.1 numerical aperture (NA)), and later was magnified for patching with a 40 × (0.8 NA) water-immersion lens fitted to an upright microscope (Nikon Eclipse E600FN; Nikon Inc, Melville NY, USA) fitted with infrared differential interference contrast optics.

Whole-cell recordings were taken from astrocytes in PFC layers II/III and V by an experimenter blinded to treatment condition. Cell bodies of astrocytes were visualized using astrocyte-specific AAVs (AAV5.*GfaABC1D*::tdTomato) that were excited by wide-field illumination (555 nm LED, Thorlabs; coupled through a 40× objective). Light power and duration were controlled via the LED driver (LEDD1B, Thorlabs, Newton, NJ, USA) by external voltage modulation. Successful patching onto astrocytes was confirmed by electrophysiological features including their hyperpolarized resting membrane potential (RMP; Saline −84.11 ± 0.294, Heroin 205—83.43 ± 0.269 mV), their linear current–voltage relationship, their inability to generate APs, and their low membrane resistance (Rm; Saline 2.7 ± 0.2955, Heroin 2.473 ± 207 0.3709 MΩ). Recordings were taken with borosilicate glass pipettes (4 to 5 MΩ) containing the following internal solution: 115 mM K-gluconate, 6 mM KCl, 5 mM glucose, 7.8 mM Na-phosphocreatine, 4 mM Mg-ATP (adenosine triphosphate), 0.4 mM Na-GTP (guanosine triphosphate) (pH 7.25 with KOH; osmolarity, 295 mml/kg. Electrophysiological signals were amplified using the Multiclamp 700B amplifier (Molecular Devices, San Jose, CA, USA), low-pass filtered at 2 kHz, and digitized at 10 kHz). Data were collected using AxoGraph X and stored for offline analysis. For the recording of the threo-beta-benzyloxyaspartate (TBOA) (non-selective blocker of excitatory amino acid transporters)- and BaCl_2_ (K^+^ channel antagonist)-sensitive currents, the extracellular solution contained antagonists of n-methyl-D-aspartate, (NMDA) receptors (D-AP5; 50 μM), ⍺-amino-3-hydroxyl-5-methyl-4-isoxazole-propionate (AMPA) receptors (NBQX; 10 μM), and γ-aminobutyric acid type A (GABAA) receptors (picrotoxin; 100 μM). 

Astrocytes were held at −80 mV, and TBOA (100 µM)- and BaCl_2_ (200 µM)-sensitive currents were evoked by single-pulse stimulation every 20 s using focal electrical stimulation with a bipolar tungsten electrode (FHC) positioned immediately medial (within ~100 μm) to the recorded astrocyte. The protocol was repeated 2 times per stimulation intensity and then averaged and analyzed. Access resistance was monitored (<20 MΩ), and recordings with an access resistance changing more than 20% between the beginning and the end of the recording were discarded. Resting membrane potential (RMP) and input resistances (Rm) were monitored for analysis of the electrophysiological properties of astrocytes. 

### 2.5. Transcardial Perfusions and Immunohistochemistry

Twenty-four hours after the final extinction session, rats were transcardially perfused with 150 mL of 0.1 M phosphate buffer (PB), followed by 200 mL of 4% *w*/*v* paraformaldehyde (PFA) in 0.1 M PB. Brains were extracted and post-fixed in PFA for 24 h. Sections (80 µm) containing the PrL cortex and NAcore were collected in 0.1 M PBS containing 0.02% sodium azide with a vibrating microtome (Leica), and stored at 4 °C until processing.

Immunohistochemistry (IHC) was performed as previously described [13,14,31]. Briefly, 3 sections per animal were blocked and permeabilized by soaking with 0.1 M PBS containing 2% (*v*/*v*) TritonX-100 (PBST) containing 2% (*v*/*v*) normal goat serum (NGS). Sections were then incubated overnight with gentle agitation at 4 °C with the following primary antibodies: Experiment 2—chicken anti-GFP (Abcam, 1:1000, ab13970 RRID:AB_300798) + mouse anti-GluA2 (Millipore, 2.13 μ/mL, MAB397 RRID:AB_11212990) or rabbit anti-Synapsin I (Abcam, 1:500, ab64581 RRID:AB_1281135), and in a separate run, rabbit anti-GFAP (Abcam, 1:1000, ab7260 RRID:AB_305808); Experiment 3—mouse anti-FLAG (Sigma, 1:2000, F1804 RRID:AB_262044) and rabbit anti-GluA1 (Abcam, 1:500, ab31232 RRID:AB_2113447); Experiment 4—rabbit anti-NeuN (Millipore1:1000, MABN140 RRID:AB_2571567). Sections were then washed 3 × 5 min with PBST, and were incubated in species-specific secondary antibodies (Invitrogen, all 1:1000) conjugated to Alexa Fluor^®^ 488 (RRID:AB_2534096, GFP), 594 (RRID:AB_2534091, FLAG), or 647 (RRID:AB_2535804, GFAP, Synapsin I, GluA1, and GluA2) diluted in PBST with 2% NGS for 5 h at room temperature (~20–25 °C) with gentle agitation. Sections were washed 3 × 5 min in PBST, then mounted on Superfrost plus slides with Prolong Gold antifade (ThermoFisher Scientific, Waltham, MA, USA). Slides were stored at 4 °C until imaging (<2 months).

### 2.6. Confocal Microscopy

For all high-resolution confocal microscopy experiments, a Leica SP8 confocal microscope (Leica Microsystems Inc., Deerfield, IL, USA) equipped with HyD detectors for enhanced sensitivity was used. Laser power, gain, and pinhole size (≤1 Airy Unit) were empirically determined to obtain optimal saturation values and subsequent deconvolution results. Laser power and gain values were then held relatively constant for the remainder of the experiment. For all experiments, Alexa^®^ 488-mediated Lck-EGFP detection was carried out using an Argon 488 nm laser line, Alexa^®^ 594-mediated FLAG detection was carried out using an OPSL 552 nm laser, and Alexa^®^ 647-mediated GluA2, GFAP, and Synapsin I detection was carried out using a Diode 638 nm laser. In Experiment 2a, fields of GFAP + cells were imaged in layers II/III and V of the PrL cortex at a 4096 × 4096 frame size, 0.1 mm Z-step size, and 1× digital zoom, with a line average of 4 using a 63× oil-immersion objective (1.4 NA). In Experiment 2b, Lck-EGFP-expressing astrocytes that were visually isolated from neighboring astrocytes in layers II/III and V as well as GluA2/Synapsin I puncta were sampled using a 63× oil-immersion objective (1.4 N.A.) at a 2048 × 2048 frame size, 0.3 mm Z-step size, and 1.5× digital zoom, with a line average of 4. Care was taken to ensure that the majority of each astrocyte in the Z-plane was imaged. In Experiment 3, apical tuft dendrites of layer V PrL-NAcore neurons were imaged as previously described [31]. Briefly, dendritic spine segments were imaged only if they met the following criteria: 1: they could be traced back to the cell body of origin, 2: they were unobscured by neighboring dendritic segments, and 3: they were past the bifurcation of the proximal dendrite (terminating in layers I and II). Dendritic spine segments (~50 mm) were collected at 1024 × 512 frame size, 0.1 mm Z-step size, and 4.1× digital zoom, with a line average of 2. Each spine segment imaged contained an immunohistochemically labeled GluA1 channel in addition to FLAG for subsequent co-registry analyses.

### 2.7. Image Analysis

All image analyses were performed by an investigator blind to experimental groups. Following acquisition, confocal Z-stacks were deconvolved using Huygens software v.2 (Scientific Volume Imaging, Hilversum, NL, USA) and then exported to Bitplane Imaris v. 9.1 (Oxford Instruments, Concord, MA, USA) for digital rendering and analysis. Complexity of GFAP arbors were analyzed as previously described [14]. Briefly, Z-stacks were cropped to 12 mm in the Z plane to identify fields of cells with homogenous signal intensity, avoiding large areas of vasculature. We used a previously published semi-automated approach to skeletonize GFAP arbors belonging to individual, non-overlapping astrocytes. In doing so, we set the minimum diameter of a GFAP filament to 1 µm. Iterative processing of GFAP skeletons were performed to accurately skeletonize each astrocyte within the field. We exported and analyzed astrocyte structure using the following outputs from the Imaris filament function: the number of astrocytes in each image, the total sum of the GFAP filament length for each cell per image, the average GFAP diameter of each cell per image, the average number of branch points for each cell per image, and the average number of 3D Sholl intersections (1 µm radius) per cell, per image. The average number of Sholl intersections was then binned by 5 µm radii from 0 to 50 µm. Data are expressed as an average for each cell per image, or an overall image average as specified in the results.

The whole-cell astrocyte synaptic contact assay was performed as described previously [13,14], with minor modifications. Briefly, confocal Z-stacks were cropped in 3D to the border of each cell to isolate each astrocyte. A 3D rendering was generated from the GFP signal from each virally labeled astrocyte. We then isolated all GFP signal to each 3D rendering to remove background. The colocalization module of Imaris was used to perform co-registry analyses as previously described. A signal intensity threshold was automatically set according to the Costes method [32] for each channel (Channel 1: GFP, Channel 2: Synapsin I or GluA2). Images that could not be assigned automatic thresholds, due to uneven signal intensity distribution through the Z-plane of the data set, were not analyzed. Voxels that contained signal above threshold for each channel were considered colocalized. The greater number of co-registered voxels indicates a greater degree of astrocyte association at synapses on a whole-cell level. In a subset of images, we also digitally modeled each Synapsin I or GluA2 puncta to determine the effect of saline, heroin + vehicle, or heroin + NAC treatments on the number of puncta within each image, normalized to the data-set volume. Data exported included the percent of the data set colocalized normalized to the saline control group (% colocalized), and the number of Synapsin I/GluA2 puncta normalized to the data-set volume.

Dendritic spine morphometric feature and GluA1 co-registry were analyzed as previously described [31]. Briefly, the filament extension of Imaris was used to manually trace each dendritic shaft and automatically label corresponding dendritic spines under input from the experimenter. Care was taken to ensure that each dendritic spine head was modeled accurately. Variables exported included the average dendritic spine head diameter (d_H_) of each segment and the dendritic spine density (spines/µm). We also binned the spine head diameters (0.2 µm bins) to generate a frequency distribution. For GluA1 co-registry analyses, a 3D rendering was then used to isolate the dendrite segment (shaft + spines), and GluA1 signal above background contained within the dendrite that was isolated for analysis. The colocalization module of Imaris was then used to generate a channel containing GluA1 and FLAG colocalization, and the ‘filament analysis’ MATLAB Xtension was used to split the channel into FLAG-GluA1 colocalization in dendritic subcompartments including the dendritic spine head and dendritic shaft for differential investigation of colocalization in respective subcompartments. Output variables included the percent of GluA1 volume normalized to the dendrite volume, the total volume of spine-head-specific GluA1 normalized to the dendrite volume, and the total volume of dendritic shaft GluA1 normalized to the dendrite volume, as well as signal intensity controls. Data are expressed as animal averages, collapsing each of the 3–5 dendrites sampled per animal into an average value of spine d_H_ and density per animal.

### 2.8. Statistical Analyses

All statistical analyses were performed with SPSS statistical software (v.28 Somer, NY, USA) or Graphpad Prism software (v. 8.0, La Jolla, CA, USA). Evoked GLT-1-dependent current from whole-cell astrocyte recordings was modeled with non-linear regression techniques, and a linear discriminant analysis with bootstrapping procedure was performed to determine if evoked GLT-dependent current was predictive of initial drug treatment. A principal components analysis was performed to produce a lower-dimensional data set before a clustering analysis was performed for all cells recorded. Self-administration and extinction data across subsequent days were compared using a mixed-model ANOVA. When analyzing three groups across a single dependent variable, a one-way ANOVA was performed followed by Tukey’s multiple comparison test if a significant effect of treatment was revealed. Normality of data within each group were analyzed using Shapiro–Wilk test. If one or more groups for each variable analyzed did not pass the Shapiro–Wilk test, a non-parametric Kruskal–Wallis test was used. In this instance, post hoc tests controlled for the false discovery rate (FDR) using the two-stage linear step-up procedure of Benjamini, Krieger, and Yekutieli if a significant effect was revealed, in which case FDR-corrected *q* values are reported. If data were sampled from a normal distribution, but between-group standard deviations were not equal, a Brown–Forsythe-corrected one-way ANOVA was performed followed by Dunnett’s T3 test when a significant effect of treatment was revealed. When three groups were compared across a repeated measure, a repeated-measure one-way ANOVA was applied followed by Tukey’s multiple comparison test comparing the mean of the repeated measure between groups. Dendritic spine frequency distribution data and dendritic spine GluA1 colocalization data were analyzed with a two-way repeated-measure ANOVA followed by Tukey’s multiple comparison test when a significant interaction was revealed. GluA1 immunoreactivity across dendritic spine head size was evaluated using linear regression, wherein all figures contain linear fits derived from the analysis. Unless otherwise stated, each data point represents an animal average derived from multiple images or SA/extinction sessions. Statistical outliers were detected according to the ROUT method (Q = 1%) and these data were removed from analyses as specified in the results. All data are expressed as the mean ± SEM, and significance was set at *p* < 0.05. A priori sample size and power estimates were derived from previously published literature [12,31].

## 3. Results

### 3.1. Experiment 1—Heroin SA and Extinction

In Experiment 1, we sought to evaluate the effects of heroin SA and extinction upon the physiological properties of PrL astrocytes using a strategy where an electrical-stimulation-induced inward current attributed to either the K^+^ or glutamate current was evaluated. The current attributed to glutamate uptake was blocked with the excitatory amino transporter inhibitor TBOA, whereas K^+^ conductance was blocked with BaCl_2_, a K^+^ channel antagonist. Of the 22 initial animals in Experiment 1, 1 animal did not finish SA due to catheter failure. Figure 2A shows a schematic of the AAV infusion in the PrL cortex and Figure 2B shows the acquisition and extinction data for yoked-saline (YS) (*N* = 11)- and heroin (*N* = 10)-treated animals. Comparing heroin and YS active lever presses across sessions, a mixed-model ANOVA reveals a significant drug-by-time interaction (F(13,221) = 3.002 *p* < 0.001). From sessions 9–14, heroin SA animals pressed the active lever significantly more than YS controls (F(1,18) = 11.30, *p* = 0.003). 

### 3.2. Experiment 1—Heroin SA and Extinction Produced Alterations in Evoked Astrocytic GLT—1 Current

To examine functional adaptations in cortical astrocytes following heroin SA and extinction, heroin-treated animals (*N* = 10) and YS-treated animals (*N* = 11) were euthanized 24 h after the final extinction session and evoked K^+^ and GLT−1-dependent currents were recorded from layer II/III and V astrocytes (N = 3–5 per animal). Figure 2C illustrates representative traces of the GLT-1 and K^+^ current in PrL astrocytes following a single stimulation in saline- and heroin-treated rats. No difference was found for the evoked K^+^ current (Figure 2D). Marginal evoked differences in the elicited GLT−1-dependent current were observed between treatment conditions, beginning at a stimulation intensity of 100 µA (Figure 2E). To further interrogate these differences, linear regression techniques were applied to model the responses following a 100 µA stimulation intensity using curve-fitting functions internal to GraphPad Prism, where it was determined that separate linear models were required to best characterize the data (F(2,26) = 84.42, *p* < 0.001, Figure 2F). In both treatment groups, model fit was evaluated using coefficients of determination (for YS R^2^ = 0.98, for heroin, R^2^ = 0.96). Next, to determine the degree to which drug treatment is predictive of the evoked GLT-1 response, the data were first bootstrapped (sample n = 1000) before a linear discriminant analysis was performed, where we show that the treatment condition can be predicted with 85.4% accuracy (R_c_ = 0.687, λ = 0.893, Wilk λ = 0.528). Obtained discriminant scores are displayed in Figure 2G, showing a clear separation between treatment conditions. To interrogate sources of variation in the evoked GLT-1-dependent current, a principal components analysis was first applied to the original data set to produce lower dimensional data while maintaining >90% of observed variance before a two-step cluster analysis was applied to the data. This analysis revealed the presence of at least two functional clusters of the PrL astrocyte response, largely independent of drug treatment (Figure 2H, silhouette coefficient > 0.5). Collectively, these results show that, while marginal, heroin SA and extinction lead to functional alterations in cortical astrocytes, though this effect may be somewhat obfuscated by drug-independent PrL astrocyte heterogeneity. Figure 2I illustrates representative viral expression in the PrL. Next, we interrogated heroin-dependent structural adaptations.

### 3.3. Experiments 2–3— Heroin Self-Administration and Extinction

Figure 3A shows self-administration and extinction data for heroin SA (*N* = 31) and YS (*N* = 20) rats used for Experiments 2–3. Of the 63 initial rats in Experiments 2–3, 12 animals did not finish SA due to catheter failure. When comparing heroin and YS active lever presses across SA sessions, a two-way repeated-measures ANOVA revealed a significant group-by-time interaction (F(13,637) = 5.112, *p* < 0.001). Heroin SA animals pressed the active lever significantly more than YS controls in sessions 9–14 (F(1,49) = 10.168, *p* = 0.002). There was no difference between groups (that were subsequently divided by treatment during extinction) in the average number of heroin infusions earned over the last 3 days of SA (Heroin + Vehicle: 29.38 ± 2.80, Heroin + NAC: 25.75 ± 4.87), indicating that the observed effects of NAC upon extinction were not dependent upon differences in initial heroin intake during SA. Figure 3B shows extinction training data for YS animals injected with vehicle (*N* = 20), and heroin SA rats injected with vehicle (*N* = 15) or NAC (*N* = 16) during the last 10 days of extinction. Figure 3C shows active lever presses averaged across the last 3 days of extinction for each group. A one-way ANOVA revealed a significant effect of treatment (F(2,48) = 12.15, *p* < 0.0001). Tukey’s multiple comparison tests revealed that heroin SA animals showed significantly more lever pressing during the last 3 days of extinction than YS rats (*p* < 0.0001) and that NAC significantly decreased lever pressing to YS values (*p* = 0.019).

### 3.4. Experiment 2A—Heroin SA and Extinction Increased Astrocyte GFAP Arbor Complexity, an Effect That Was Prevented by Repeated NAC Treatment during Extinction

Here, we used an astrocyte-specific membrane-targeted EGFP reporter virus to label the astrocyte plasma membrane (AAV5-*GfaABC1D*::Lck-EGFP) to evaluate GFAP arbor complexity following heroin SA, extinction, and NAC treatment. Figure 4 shows a representative immunohistochemical preparation indicating that EGFP expression is indeed specific to GFAP+ astrocytes. Of the initial 10 YS rats used in experiment 2A, 2 were not included due to technical issues related to IHC processing, 2 Heroin + Vehicle rats were not included due to technical issues related to IHC, and 1 Heroin + NAC animal was excluded due to lack of analyzable images. Figure 5A shows a representative field of GFAP+ cells in the PrL cortex (left); the same image is shown with an overlaid Imaris-based skeletonization (white) of each GFAP+ cell (middle), and an individual GFAP+ cell (yellow) isolated from the other cells (right). Using this high-throughput approach, we analyzed GFAP complexity in three to seven Z-series data sets from eight YS rats, seven Heroin + Vehicle rats, and nine Heroin + NAC rats using several different metrics. We first analyzed the average number of GFAP+ cells per field. A one-way ANOVA revealed no effect of treatment (F(2,92) = 1.23, *p* = 0.30, Figure 5B). For complexity analyses, we first analyzed the total sum of GFAP filament length within each Z-series data set (Figure 5C). A Shapiro–Wilk test indicated that data in the YS (W = 0.9, *p* = 0.0005) and Heroin + NAC (W = 0.94, *p* = 0.04) groups were non-normally distributed. Accordingly, a Kruskal–Wallis non-parametric ANOVA was used, which indicated a significant difference between groups (*H* = 12.99, *p* = 0.002). The two-stage linear step-up procedure revealed that heroin increased total GFAP filament length per image when compared to YS (*q* = 0.0007), which was normalized by NAC treatment during extinction (*q* = 0.004). We next analyzed the average GFAP filament diameter for each cell per Z-series data set (Figure 5D). One Z-series data set from YS and two Z-series data sets from the Heroin + NAC group were identified as statistical outliers. A Brown–Forsythe test revealed that the standard deviations were significantly different between groups (F(2,89) = 3.26, *p* = 0.04). As such, these data were analyzed using a Brown–Forsythe-corrected ANOVA which revealed there was no significant effect of treatment (F(2,74.38) = 0.626, *p* = 0.54). We next analyzed the average number of branch points of each cell, within each Z-series data set (Figure 5E). One Z-series data set from the YS group was identified as a statistical outlier. A Shapiro–Wilk test indicated that data from the YS group were non-normally distributed. Accordingly, a Kruskal–Wallis test was used which revealed a significant difference between groups (*H* = 14.61, *p* = 0.0007). The two-stage linear step-up procedure revealed that heroin increased the average number of branch points within GFAP filament arbors compared to YS (*q* = 0.0002) which was normalized by NAC treatment during extinction (*q* = 0.003). We also performed a 3D Sholl analysis for each cell within each image using 1 µm 3D concentric sphere radii (which were then compiled into 5 µm bins) and analyzed the average number of intersections as a function of distance from the center of each cell (Figure 5F). A repeated-measures one-way ANOVA revealed a significant difference between group means (F(2,20) = 19.23, *p* < 0.0001). Tukey’s multiple comparison test revealed that heroin increased the average number of Sholl intersections compared to YS (*p* = 0.0002), which was normalized by NAC treatment during extinction (*p* < 0.0001). In a subsequent analysis, we focused on the first 15 µm 3D sphere radius from the centroid of each cell to investigate the primary GFAP+ branches (Figure 5G). In these data, a one-way ANOVA revealed a significant difference between groups (F(2,92) = 8.41, *p* = 0.0004). Tukey’s multiple comparison test revealed that heroin increased the number of Sholl intersections in this region when compared to YS (*p* = 0.003), which was normalized by NAC treatment during extinction (*p* = 0.0007). Figure 5H shows an individual cell with a representative skeletonized GFAP arbor from each treatment group.

### 3.5. Experiment 2B—Heroin SA and Extinction Increased Astrocyte Association with Multiple Synaptic Markers, an Effect That Was Prevented by Repeated NAC Treatment during Extinction

We performed two separate astrocyte–synaptic contact analyses using PrL cortical sections collected from the three experimental groups, YS (*N* = 10), Heroin + Vehicle (*N* = 9), and Heroin + NAC (*N* = 10), to examine the association of astrocytes with synaptic markers following heroin SA, extinction, and NAC treatment. In the first set, the astrocyte structure was visualized with viral-vector-mediated membrane-targeted EGFP expression achieved with the AAV5.*GfaABC1D*::Lck-EGFP viral vector and further amplified with GFP IHC whereas excitatory synapses were labeled with GluA2 IHC as described previously [31]. The second set of astrocyte–synaptic contact analyses was performed identically, except that synapsin I IHC was used as a general synaptic marker [13]. We imaged an average of 3–5 cells/animal for both GluA2 and synapsin I analyses. For the GluA2 analyses, the following were excluded: one YS animal for the missing localization of the Lck-EGFP virus expression, one YS animal for the lack of cells satisfying the Costes threshold assignment, one YS animal for issues related to IHC, one Heroin + Vehicle animal which had insufficient Lck-EGFP virus expression, one Heroin + NAC animal for the lack of cells satisfying the Costes threshold assignment, and one Heroin + NAC which had no more available tissue.

Figure 6A shows a representative PrL cortical Lck-EGFP-labeled astrocyte (green) and punctate signal corresponding to GluA2 (red). To aid in the visualization of the punctate GluA2 synaptic marker signal, an individual optical section from the many images that comprise each Z-series data set is shown. The left panel shows the raw signal, the middle panel shows the GluA2 signal contained within the 3D surface of the astrocyte membrane, and the right panel shows colocalized voxels. Within each panel, a smaller depiction of the full Z-series data set is visible in the top right corner, with an inset depicting the field seen within the single optical section view. Figure 6B shows the percent of the co-registry of the GFP and synaptic marker signals across the three treatment groups, a metric of astrocyte–synaptic interaction [13,14,33,34]. Since our experimental measurements utilized both a general synaptic marker and an excitatory synaptic marker, all data were normalized to YS animals and expressed as % of control. This was done for ease of comparison between markers and cohorts of animals. However, it is important to note that, although baseline values differed slightly from cohort to cohort, the effect of heroin was reproducible across both cohorts (Figure 7A–D). For GluA2, a Brown–Forsythe-corrected ANOVA revealed a significant effect of treatment (F(2,9.743) = 4.74, *p* < 0.01). The two-stage linear step-up procedure indicated that the Heroin + Vehicle group showed an increased co-registry of GluA2 and GFP compared to the YS group (*p* = 0.01), which was normalized by repeated NAC treatment during extinction (*p* < 0.02, Figure 6B). We next determined whether this effect was due to an increased number of GluA2 puncta within each data set. A one-way ANOVA revealed no significant effect of treatment (F(2,20) = 2.24, *p* = 0.13, Figure 6C). Figure 6D shows representative 3D-reconstructed astrocyte for each treatment group with GluA2 puncta (top) as well as the extent of synaptic contact as visualized by the white GFP and GluA2 co-registered voxels (bottom).

As described above, we also performed the same imaging and analysis for Lck-EGFP and Synapsin I, using tissue from the same animals to determine whether the observed effect was specific to synapses containing the GluA2 subunit of AMPA receptors. One YS, two Heroin + Vehicle animals, and two Heroin + NAC animals were removed from the analysis for the following reasons: YS + Vehicle: off-target Lck-EGFP virus expression (*N* = 1); Heroin + Vehicle: lack of available tissue (*N* = 1) or insufficient Lck-EGFP virus expression (*N* = 1); Heroin + NAC: lack of a sufficient number of analyzable cells after imaging (*N* = 1); and technical issues related to IHC (*N* = 1). A one-way ANOVA revealed a significant main effect of treatment (F(2,21) = 6.51; *p* = 0.006; Figure 6E). Tukey’s multiple comparison test revealed that heroin SA and extinction increased astrocyte–synaptic interaction as assayed by Lck-EGFP-Synapsin I co-registration when compared to YS controls (*p* = 0.012), which was normalized by NAC treatment during extinction (*p* = 0.012). Akin to analyses performed for GluA2, we also analyzed the number of Synapsin I voxels in a subset of images from each animal (Figure 6F). A Shapiro–Wilk test indicated that data from YS controls (W = 0.83; *p* = 0.04) were non-normally distributed. A Kruskal–Wallis test indicated no difference between groups (*H* = 2.04; *p* = 0.36). There was no difference between groups in astrocyte complexity as measured by the ratio of the surface area to volume for cells used in either the GluA2 or Synapsin I data sets (Figure 7E,F).

### 3.6. Experiment 3A—Heroin SA and Extinction Impacts Spine Density and Spine Head Diameter on PrL-NAcore Neurons, an Effect That Was Partially Reversed by Repeated NAC Treatment during Extinction

Others have shown that heroin SA decreases the number of dendritic spines on PrL cortical neurons after one month of abstinence [35]. Here, we sought to determine whether heroin SA and extinction altered dendritic spine morphometrics specifically in PrL cortical neurons projecting to the NAcore. We imaged 5–8 dendritic spine segments from YS (*N* = 10), Heroin + Vehicle (*N* = 6), and Heroin + NAC (*N* = 6) animals after extinction. Figure 8A shows a representative layer V PrL-NAcore neuron fully labeled with the highlighted area indicating the region sampled at the apical tuft. Figure 8B shows the dendritic spine segment sampled from the highlighted region in Figure 8A. Figure 8C shows the 3D rendering of the processed dendrite for spine detection and Figure 8D shows a color-coded map of the dendritic spine head diameter (d_H_) with warmer colors indicating larger values. We first analyzed the average dendritic spine d_H_ across the three treatment groups. A one-way ANOVA revealed a significant effect of heroin treatment (F(2,19) = 20.70, *p* < 0.0001, Figure 8E). Tukey’s multiple comparison test revealed that heroin SA and extinction increased the average dendritic spine d_H_ (*p* < 0.0001), which was not altered by NAC treatment during extinction. Heroin + NAC treatment likewise increased d_H_ relative to saline-treated animals (*p* = 0.0007). Next, we investigated whether this effect was specific to smaller or larger spines. To do so, we binned the spine d_H_ (0.2 µm bins) and generated a frequency distribution. A two-way repeated-measures ANOVA revealed a significant treatment-by-bin interaction (F(8,76) = 8.144, *p* < 0.0001, Figure 8F). Tukey’s multiple comparison test revealed that heroin SA and extinction, regardless of NAC treatment, decreased the proportion of smaller spines (<0.2—*p* < 0.0001; 0.2–0.4—*p* < 0.05), but increased the proportion of spines with a larger d_H_ (0.4–0.6—*p* < 0.01) compared to YS rats. Interestingly, the proportion of spines with a d_H_ between 0.6–0.8 was increased in the Heroin SA + Vehicle group (*p* = 0.008), but there was no effect of NAC on this specific bin compared to the YS treatment group (*p* = 0.17). We next analyzed the average dendritic spine density in all three treatment groups from the same dendrite segments. A one-way ANOVA revealed a significant effect of treatment (F(2,19) = 13.46, *p* = 0.0002, Figure 8G). Tukey’s multiple comparison test indicated that heroin SA and extinction decreased the average spine density compared to saline controls (*p* = 0.0003), which was prevented by NAC treatment during extinction (*p* = 0.002). To determine which spine head diameter bins accounted for the altered spine density, we performed a similar analysis as in Figure 8F, but instead normalized the number of spines within each bin to the length of the dendrite. A two-way repeated-measures ANOVA revealed a significant treatment-by-bin interaction (F(8.76) = 8.722, *p* < 0.0001, Figure 8H). Tukey’s multiple comparison test indicated that heroin SA and extinction (*p* < 0.0001), regardless of NAC treatment (*p* = 0.0003), decreased the density of spines with a head diameter less than 0.2 µm. However, the Heroin + Vehicle group showed decreased spines in the 0.2–0.4 µm bin compared to YS (*p* < 0.0001), which was normalized in the Heroin + NAC group (*p* < 0.0001 compared to Heroin + Vehicle). Interestingly, the density of larger spines with head diameters between 0.4–0.6 µm were unaffected in the Heroin + Vehicle group compared to YS (*p* = 0.194), yet Heroin + NAC increased the density of these spines compared to the YS (*p* = 0.0125) and Heroin + Vehicle (*p* = 0.0002) groups. These effects on dendritic spine morphometric properties were likely not due to inherent differences in the FLAG signal intensity within the spine heads as there was no difference between groups in the average FLAG intensity within the spine head region when normalized to saline controls (Figure 9A). Representative spine segments (15 µm cropped segments) are shown in Figure 8I.

### 3.7. Experiment 3B—Heroin SA and Extinction Increased GluA1 Immunoreactivity in Dendritic Shafts and Spine Heads of PrL-NAcore Neurons, and NAC Potentiated Spine-Head-Specific GluA1

We previously showed that cocaine SA followed by one week of abstinence increases dendritic spine head diameter in the apical tuft dendrites of PrL-NAcore neurons. We also demonstrated that cocaine abstinence-duration-dependent structural plasticity occurs concomitantly with alterations in GluA1 immunoreactivity within the dendritic shaft and at dendritic spine heads. Here, we investigated whether the adaptations in dendritic spine morphometrics following the extinction of heroin SA (with or without repeated NAC treatment) were associated with altered GluA1 immunoreactivity in the discrete dendritic subcompartments of the PrL-NAcore apical tuft dendrites described above. When analyzing the total volume of GluA1 immunoreactivity in the dendritic spine heads and shaft combined, relative to the total dendrite volume, a one-way ANOVA revealed an omnibus significant effect of NAC treatment upon GluA1 immunoreactivity (F(2,19) = 10.56, *p* = 0.0008). Tukey’s multiple comparison test indicated that the Heroin + Vehicle (*p* = 0.02) and Heroin + NAC (*p* = 0.0009) groups displayed increased GluA1 immunoreactivity compared to saline controls (Figure 10A). In the case of the Heroin + NAC, but not Heroin + Vehicle, group, increases in GluA1 volume could occur in part due to increased overall GluA1 signal intensity, as the average GluA1 intensity (spine heads + dendritic shaft) was significantly higher in this group (main effect of treatment: (F(2,19) = 4.27, *p* = 0.029, Tukey’s post hoc comparing Heroin + NAC to Heroin + Vehicle—*p* = 0.025, Figure 9B).

Next, we focused on dendritic-spine-head-specific GluA1 immunoreactivity. Here the volume of the signal in spine heads was expressed relative to the total dendrite volume, as a function of treatment. A Brown–Forsythe ANOVA revealed a significant effect of treatment (F(2,9.028) = 20.22, *p* = 0.0005). Dunnett’s T3 multiple comparison test revealed that the Heroin + Vehicle (*p* = 0.03) and Heroin + NAC (*p* = 0.004) groups showed increased spine-head-specific GluA1 signal accumulation when compared to the Saline + Vehicle group (Figure 10B). However, in this instance, there was no difference between groups in the average GluA1 intensity in the co-registered volume (Figure 9C).

Finally, we analyzed dendritic-shaft-specific GluA1 immunoreactivity, relative to the total dendrite volume, as a function of treatment. A one-way ANOVA revealed a significant effect of treatment (F(2,19) = 7.73, *p* = 0.004). Tukey’s multiple comparison test indicated that the Heroin + Vehicle (*p* = 0.033) and Heroin + NAC (*p* = 0.005) groups showed increased dendritic-shaft-specific GluA1 immunoreactivity compared to the Saline + Vehicle group (Figure 10C). The Heroin + NAC group displayed increased GluA1 intensity in the dendritic shaft subcompartment compared to both the Saline + Vehicle and Heroin + Vehicle groups (Figure 9D). This is to be expected given the results discussed above, as the shaft region constitutes a large portion of the total dendritic segment volume. Interestingly, correlation analyses revealed that % GluA1 volume in the total segment normalized to the total dendrite volume (y = 34.10X − 8.093, R *=* 0.71, R^2^ = 0.51, *p* = 0.0002, Figure 10D), % GluA1 volume in the spine heads normalized to the total dendrite volume (y = 4.457X − 1.231, *R =* 0.55, *R*^2^ = 0.300, *p* = 0.008, Figure 10E), and % GluA1 volume in the shaft normalized to the total dendrite volume (y = 29.65X − 6.853, R = 0.71, R^2^ = 0.504, *p* = 0.0002, Figure 10F) showed a positive correlation with changes in the spine head diameter as a function of treatment. In contrast, the % GluA1 volume in the total dendrite, spine heads only, and shaft only did not correlate with changes in dendritic spine density as a function of treatment (Figure 11A–C). Representative dendritic spine segments with GluA1 immunoreactivity in the two subcompartments are shown in Figure 10G. 

### 3.8. Experiment 4—Both NAC Treatment and Delivery of shTSP-2 to PrL Astrocytes Blunt Cue-Induced Relapse to Heroin Seeking 

Apart from the well-documented ability of NAC to normalize glutamate homeostasis [10,15,24], our data demonstrate that NAC treatment reversed heroin-mediated reductions in spine density at PrL-NAcore neurons. Accordingly, we hypothesized that NAC may exert its therapeutic effects through the restoration of dendritic spine populations at key subsets of cortical neurons following heroin SA. Astrocyte release of both TSP 1 and 2 mediate neuronal synaptogenesis, with astrocytic TSP signaling also required for the formation of silent synapses following cocaine exposure [30]. We thus hypothesized that the genetic elimination of TSP-2, and loss of an important signal for synaptogenesis, would disrupt the therapeutic efficacy of NAC treatment. To test the hypothesis that the delivery of shTSP-2 might blunt the effects of NAC treatment during extinction, rats received an infusion of either AAV5.*GfaABC1D*::Lck-EGFP (control, *N* = 11) or AAV5.*GfaABC1D*::shTSP2-GFP (shTSP, *N* = 14), an shRNA targeted to TSP-2 [30]. Figure 12A depicts a schematic of the AAV infusion and Figure 12B displays a schematic of viral placements in addition to representative low- and higher-magnification images illustrating isolation of expression of the viral constructs to astrocytes. Rats then underwent heroin SA and extinction (Figure 12C,D). During the last 10 days of extinction, rats received injections of either vehicle or NAC. Following extinction, rats underwent cued-relapse testing for heroin. PrL delivery of shTSP-2 was not found to alter heroin SA or extinction active lever pressing (Figure 12C,D). A 2 × 2 ANOVA revealed that both NAC treatment (F(1,23) = 7.480, *p* = 0.013) and the delivery of shTSP-2 (F(1,23) = 5.005, *p* = 0.037) significantly reduced active lever pressing during the last three days of extinction. Though NAC+ shTSP-2 treatment did result in a marginal decrease in active lever pressing during the last three days of extinction, the effect was not statistically significant. Similarly, both NAC (F(1,23) = 6.155, *p* = 0.02) and the delivery of shTSP-2 (F(1,23) = 4.423, *p* = 0.04) significantly blunted cued-relapse to heroin seeking. Similar to findings from extinction data, NAC + shTSP-2 treatment resulted in a marginal decrease in cued-relapse active lever pressing, though the effect was not statistically significant compared with either the Lck+ NAC or shTSP2 + vehicle treatment groups (Figure 12E).

## 4. Discussion

Here, we show that heroin SA and extinction induce functional adaptations in cortical astrocytes. Our data demonstrate that heroin SA followed by extinction enhances the fundamental cytoskeletal cellular complexity of astrocytes in the PrL cortex, a finding that was accompanied by the increased association of membranous astrocytic processes, virally labeled with membrane-targeted EGFP expression, with two distinct synaptic markers. Both of these effects, as well as heroin seeking during extinction, were normalized by repeated NAC treatment during the last 10 days of extinction training, confirming that NAC reduces responding during extinction as others have reported [19]. Apart from impacts on cortical astrocytes, heroin SA and extinction also increased the average spine d_H_ on apical tuft dendrites of PrL cortical neurons that project to the NAcore, an effect that was accompanied by decreased spine density. Interestingly, NAC treatment during extinction normalized heroin-induced decreases in dendritic spine density but did not impact dendritic spine d_H_. Moreover, NAC treatment was shown to block cue-induced relapse to heroin. These results suggest that the ability of NAC to prevent heroin relapse in preclinical models may not only be due to its action in the NAcore but also include actions that normalize structural plasticity in astrocytes, as well as neurons within the PrL cortex that project to the NAcore. Further, our results indicate that shTSP-2 delivery to cortical astrocytes likewise decreases cue-induced reinstatement to heroin, thereby cementing the role of PrL astrocytes in the cycle of relapse to heroin addiction. 

### 4.1. Heroin SA and Extinction Evoke Functional Alterations in PrL Astrocytes

Our data are among the first to show that heroin SA and extinction evoke functional alterations in evoked GLT-1-dependent current in cortical astrocytes, a finding that is consistent with our presently reported structural adaptations following heroin exposure. While the magnitude of these functional alterations is somewhat small, here, we show that these alterations can be used to predict prior heroin exposure with ~85% accuracy. The relevance of these alterations to cue-induced heroin reinstatement is an important future direction. 

### 4.2. Evoked GLT-1-Dependent Current Reveals a Functional Clustering of Cortical Astrocytes, Independent of Heroin Treatment

Interestingly, our data provide evidence of a clustering of astrocyte subpopulations in the PrL that display characteristic functional properties that appear to occur independent of drug treatment. This finding mirrors existing data to support the existence of diverse heterogeneity in astrocyte populations. Despite being the most abundant cell type in the adult brain, early classification schemes of astrocyte subpopulations are inconsistent with emerging transcriptomic and morphological data that collectively suggest an incredible degree of diversity in astrocyte populations. Indeed, recent transcriptomic investigations have revealed considerable heterogeneity among astrocyte populations both across brain regions and within a single region, including the cortex [36]. Similarly, others have reported astrocyte-specific receptor heterogeneity across regions of the brain such as the thalamus [37]. Importantly, this diverse heterogeneity is maintained not only in healthy brains, but also across disease states [38]. The exact contributions of astrocyte heterogeneity to motivational and cognitive processes remain elusive, however. Given the distribution of astrocytes throughout the brain and their central importance to global neural functioning, it is clear that future studies of astrocyte contributions to heroin SA and extinction should include investigations into cell heterogeneity in an attempt to elucidate the role of these unique subpopulations of astrocytes. Moreover, the role that this functional heterogeneity plays in cue-induced relapse remains to be explored.

### 4.3. Repeated NAC Treatment Facilitated Extinction Training

Our data provide the first preclinical evidence for NAC acting to augment structural plasticity in a cortical region; we suspect that these adaptations play a role in NAC’s ability to prevent cue-induced reinstatement. Although NAC’s ability to prevent relapse has yet to be tested in the clinic for opiate use disorder, preclinical data indicate that daily NAC treatment during extinction reduces extinction responding as well as cue- and heroin-induced drug seeking for up to 40 days after the last injection [19]. Ultimately, NAC has shown promise preclinically in decreasing relapse for several drugs of abuse [14,20,22,23,24]. However, it is critical to note that NAC’s ability to suppress relapse clinically, at least in the case of cocaine, is minimal to date [39]. The general consensus is that NAC is effective in increasing the abstinence duration in patients who are already abstinent but fails to decrease relapse rates in active users [39,40,41]. Despite its modest clinical efficacy, its well-described mechanism of action and ability to target drug-induced adaptations in non-neuronal cells substantiates its role as a powerful research tool for understanding astrocytic dysfunction in addiction.

### 4.4. Repeated NAC Treatment during Extinction Reversed Increases in Astrocyte Complexity and Association of PAPs with Synapses in PrL Cortex Induced by Heroin SA and Extinction

Here, we provide evidence that extinction from heroin SA increased GFAP cytoskeletal complexity without altering the number of GFAP+ cells in the PrL cortex. We conclude that this is not due to a transition of astrocytes to a reactive state as there was no enhancement of average GFAP filament diameter across groups, a phenomenon that serves as a canonical index of reactive astrocytes [42]. Given the profound impacts observed in PrL cortical astrocytes, the signal transduction required for this event is an immediately apparent question. It is possible that this effect may be due to heroin’s direct impact on opioid receptors expressed on astrocytes. mRNA for all three opioid receptor subtypes is present in astrocytes, and PrL cortical astrocytes in particular express an abundance of mu opioid receptor mRNA [43], the primary receptor engaged by morphine [44]. Further, the incubation of cortical astrocyte cultures with morphine enhances intracellular Ca^2+^ signaling [45], resulting in an increased area occupied by astrocytes [46]. In addition, astrocytic mu opioid receptor expression has been directly linked to the rewarding properties of drugs of abuse; e.g., activation of mu opioid receptors on hippocampal astrocytes induces conditioned place preference [47]. Thus, it is likely that one component of the observed increase in GFAP complexity in the PrL cortex of heroin-exposed rats is due to the chronic tone on astrocytic mu opioid receptors. In support, a study performed in the early 1990s showed that cultured astrocytes treated with morphine increase their branching complexity, yet morphine decreased astrocyte proliferation [27]. Alternatively, chronic injections of morphine in rats have been shown to increase GFAP protein expression in the frontal cortex [26], yet this study did not have a prolonged withdrawal period. While we did not causally link increased GFAP complexity to relapse behavior, repeated NAC treatment prevented the action of heroin at cortical astrocyte GFAP arbors and is known to reverse drug-induced deficits in astrocyte function linked to relapse propensity [48,49]. Whether normalized cortical GFAP complexity is required for NAC’s ability to suppress relapse remains to be determined. However, as an important caveat, future examinations should likewise consider astrocyte maturation in the context of these manipulations, which has been linked to enhanced GFAP arbor complexity at key developmental timepoints also associated with synaptogenesis [50,51].

Extinction from cocaine [13,34], heroin [12], or methamphetamine [14] SA leads to the decreased association of PAPs with synaptic markers in the NAcore. This finding can be interpreted as a withdrawal of PAPs, which are highly enriched in GLT-1 [52], from synapses [53,54]. Indeed, ezrin, a protein that links the astrocyte actin cytoskeleton with the astrocyte membrane and is highly enriched on peripheral astrocytic processes, has been implicated in this process. Specifically, knockdown of NAcore ezrin has been previously demonstrated to produce a similar withdrawal of PAPs from synapses comparable to the magnitude of withdrawal following extinction training [12]. In this same study, ezrin knockdown prevented the reassociation of PAPs with synaptic markers that occurs following cued-relapse and perpetuated cue-induced drug seeking [12]. This connection between astrocyte structure and function is readily made as drug-induced retraction of glial processes in the NAcore is often paired with decreased GLT-1 expression and reduced glutamate clearance [55]. While the specific mechanisms behind the downregulation of GLT-1 in the NAcore following drug SA remains elusive, possible candidates include epigenetic mechanisms [56] and potentially a disruption in GLT-1 recycling between the astrocyte plasma membrane and intracellular compartments, though more directed experimentation is required to confirm the latter hypothesis. We show here that extinction from heroin SA leads to the increased association of PAPs with synaptic markers in the PrL cortex, an effect that was specific to heroin and was not observed following the extinction of cocaine SA [34]. While further investigation would be required to determine if heroin SA and extinction impacts cortical glutamate clearance, we would not predict reduced glutamate clearance, as astrocyte retraction from synapses was not observed. The GFAP cytoskeleton represents a long-standing canonical maker for astrocytes and has been often utilized as a means to interrogate the astrocyte structure [57,58] and activation states [59]. However, it of course does not accurately recapitulate the membranous extremities of astrocytes [60] that extend beyond the GFAP skeleton. Accordingly, alterations in the cytoskeletal properties of astrocytes at the level of GFAP arbors do not necessarily mandate that adaptations in the astrocyte–synaptic interaction will be observed. As an example, methamphetamine SA and extinction decreased the PAP–synapse association, but not GFAP cytoskeletal complexity, in the NAcore [14]. Here, we do observe that heroin-induced increases in GFAP complexity were accompanied by a ~75% increase in the association of PAPs with synaptic markers in the PrL cortex following extinction from heroin SA. Interestingly, both adaptations in GFAP structure and astrocyte–synaptic interaction were normalized by repeated NAC treatment. One potential alternative explanation for our observations is that there was increased synaptogenesis in the PrL cortex after the extinction of heroin SA, with no alteration in the motility of astrocytes. However, this is unlikely given that we did not observe any difference between groups in the number of GluA2 or Synapsin I puncta.

It has been estimated that a single astrocyte makes contact with ~100 dendrites and ~100,000 synapses in the rodent brain [61]. Our astrocyte–synaptic co-registration data could be interpreted either as each astrocyte contacting a greater number of synapses in the PrL cortex after extinction or each astrocyte may be embracing individual synapses to a greater extent, though more sophisticated microscopy techniques such as electron microscopy would be required to address this issue. Additionally, some combination of these scenarios may also be occurring. Each of these potential outcomes would produce an overall increase in the co-registry of astrocyte membrane and synaptic marker signals. Astrocyte ensheathment of synapses shapes excitatory transmission by establishing the three-dimensional space in which neurotransmission occurs. More specifically, astrocyte ensheathment of synapses is a dynamic process that regulates the profiles of synaptic and non-synaptic receptor activation, allowing for astrocytic tuning of synaptic communication on a synapse-by-synapse basis. This modality of regulation has been directly explored by studies that examine how the degree to which astrocytes associate and physically interact with synapses is shaped by neuronal activity [62,63]. As an example, whisker stimulation has been shown to increase GLT-1 expression in the barrel cortex, which is associated with increased ensheathment of PAPs at enlarged dendritic spine heads [64]. Additionally, cue-induced reinstatement of heroin seeking has been associated with dynamic changes in the PAP–synapse association in the NAcore during the course of the reinstatement session that was directly linked to relapse behavior [12]. Together. these data demonstrate that astrocytes physically respond to changes in neurotransmission with physiologically and behaviorally relevant consequences. Moreover, the fidelity of synaptic transmission is tightly controlled by the PAP insulation of synapses; too little association is linked to glutamate overflow [12,13,14] whereas too much insulation can actually impede AMPA receptor-mediated excitatory transmission through enhanced glutamate uptake [65], a means of astrocyte regulation of synaptic function. Our data thus contribute to the evolving understanding of the astrocytic regulation of neuronal function, which was initially postulated to occur during sleep by Cajal almost 150 years ago [66]. Further experimentation will be needed to determine where astrocytic synapse ensheathment in the PrL cortex from heroin-experienced animals lies on this functional continuum. Whether PAP association with synapses has reached a “ceiling” and is impeding AMPA receptor transmission or whether the increased association leads to a suppression of volume transmission and a solidification of discrete excitatory inputs in a supraphysiological manner is yet to be determined.

### 4.5. Extinction from Heroin SA Altered Dendritic Spine Morphological Profiles and GluA1 Immunoreactivity in Discrete Dendritic Subcompartments of Prelimbic Cortical Neurons Projecting to the Nucleus Accumbens Core

Using similar viral-vector-based pathway-specific labeling of dendritic spines, we previously found that cocaine SA followed by one week of abstinence decreased the density of apical tuft dendritic spines of PrL-NAcore neurons, yet the remaining spines were enlarged and concomitantly displayed increased GluA1 and GluA2 immunoreactivity in the spine heads [31]. A previous study found that heroin SA and one month of abstinence decreased the density of dendritic spines of the PrL cortical layer V pyramidal neurons, with no reported alterations in the morphometric properties of the spines themselves [35]. To accompany the astrocyte–synapse interaction data discussed above, we investigated potential alterations in dendritic spine morphological profiles in PrL cortical neurons projecting to the NAcore. Akin to one week of abstinence from cocaine SA, we discovered that extinction from heroin SA decreased the density of dendritic spines along dendrites at the apical tuft of PrL-NAcore neurons, yet these neurons showed an increase in the average spine head size, as well as elevated GluA1 co-registry in the dendritic spine heads. These data are largely consistent with studies demonstrating that repeated treatment of cultured cortical neurons with morphine leads to destabilization of pre-existing spines [67]. In our data set, heroin-mediated decreases in dendritic spine density were prevented by repeated NAC treatment; yet the heroin-evoked increases in spine head diameter were not. In fact, to our surprise, we found that NAC-treated animals showed a potentiation of GluA1 immunoreactivity in dendritic spine heads relative to Heroin + Vehicle animals. Generally, an increase in the number of spines, the average spine head size, as well as AMPA receptor occupancy in spine heads, are interpreted as evidence for synaptic potentiation [68,69,70,71]. Thus, this finding is initially counter-intuitive given that NAC normalized the decrease in density but NAC-treated animals retained enlarged dendritic spines following heroin SA and extinction, suggesting that NAC is driving synaptic potentiation in the PrL-NAcore circuit. However, it should be noted that the apical tuft is a site of convergence of glutamatergic inputs from midline thalamic nuclei as well as dopamine afferents from the ventral tegmental area [72,73,74]. Thus, it remains possible that heroin-induced adaptations in spine morphology and AMPA receptor accumulation at this particular part of the dendritic arbor might not translate into whole-cell alterations in physiological indices of potentiation. Rather, this may reflect the potentiation of discrete inputs onto PrL-NAcore neurons which may or not may be counteracted by alterations in the synaptic strength of alternative inputs at other parts of the dendritic arbor. Future studies should examine whether heroin SA and extinction alter the electrophysiological properties of PrL-NAcore neurons and whether repeated NAC treatment during extinction has an impact on such alterations. Regardless, given that NAC is known to reverse drug-induced deficits in astrocyte function [18], we suspect that NAC’s ability to restore dendritic spine density following heroin SA and extinction is mediated by augmented astrocyte function, perhaps through altering the release patterns of astrocyte-derived neuromodulators that regulate synaptogenesis.

### 4.6. NAC Treatment and Downregulation of Astrocyte TSP-2 Decreased Cued-Relapse to Heroin Seeking

NAC treatment and delivery of shTSP-2 on PrL astrocytes independently of NAC treatment decreased cued-relapse to heroin seeking. These novel findings extend our finding that NAC treatment blunted extinction responding following SA, and further suggest a therapeutic role for NAC in opioid use disorder. Although, as described previously, findings suggesting NAC’s ability to suppress relapse in a clinical setting are minimal [18,39], and few, if any, existing studies directly investigated NAC’s ability to block cued-relapse to heroin seeking. As discussed above, we predicted that NAC’s ability to restore dendritic spine density following heroin SA and extinction may be mediated by augmented astrocytic function and the differential release of the astrocytic regulators of synaptogenesis. Thus, we initially hypothesized that the downregulation of astrocytic TSP-2 on PrL astrocytes would blunt the ability of NAC to both facilitate extinction training and to reduce cued-relapse following heroin SA. However, this was not the case. Despite this result, we did observe that the delivery of shTSP-2 to cortical astrocytes was sufficient to blunt cue-induced heroin seeking.

It is well-documented that astrocytes release a variety of neurotransmitters, neuromodulators, and growth factors that can have a profound impact on neuronal plasticity, (for review, see [75,76]). As an example, the astrocyte release of thrombospondin-1 (TSP—1) activates α2δ-1 receptors expressed on neurons to induce spinogenesis and excitatory synapse formation [77]. Dysfunction of this system has been linked to reduced dendritic spine density in Down Syndrome models [78], as well as elevated dendritic spine density in the NAc of animals receiving chronic cocaine injections [30]. Furthermore, the loss of astrocyte TSP-2 has been demonstrated to block the generation of silent synapses following cocaine training [30]. Presently, we report that the delivery of shTSP-2 to PrL astrocytes, independent of NAC treatment, is likewise sufficient to decrease cue-induced heroin relapse, which likewise suggests that cue-induced heroin relapse requires PrL astrocyte-mediated synaptogenesis, though more experimentation is required to directly test this hypothesis. Here, given that shTSP2 treatment did not block the ability of NAC to blunt cue-induced heroin relapse, an important alternative interpretation is that NAC may be protective with respect to heroin-induced reductions in PrL-NAcore dendritic spine density as opposed to our original hypothesis that NAC functions via synaptogenesis. However, additional experimentation is required to determine this distinction, and this research represents an important future direction of the present experiments.

It is possible that the heroin-induced decrease in spine density could be linked to dysfunctional gliotransmission following extinction, and that NAC’s ability to drive spinogenesis could be linked to the normalization of such dysfunction. For example, repeated morphine administration in mice downregulates TSP expression in the cortex 24 h after the final injection, as well as in cultured astrocytes following repeated morphine treatment [79]. One mechanism by which heroin may impact TSP expression is through increased oxidative stress, as evidenced by the induction of oxidative stress in humans following chronic heroin exposure [80] and also in cultured cortical neurons [81,82]. Moreover, cultured primary astrocytes reduce TSP-1 gene expression under increased oxidative stress conditions, which is prevented by NAC [83]. With these considerations in mind, we were presently surprised that the delivery of shTSP-2 to PrL astrocytes did not block the ability of NAC to reduce both the extinction and cued-relapse to heroin, suggesting that their effects are mediated by independent mechanisms. It could be the case, however, that heroin exposure uniquely engages TSP-1 versus TSP-2. Echoing this hypothesis, it has previously been reported that cocaine administration elevates the presence of silent synapses in both wild-type and TSP-1 knockout mice, though cocaine administration was not shown to increase silent synapse formation in TSP-2 knockout mice [30].

Given that one well-documented mechanism by which NAC counteracts oxidative stress is increased glutathione synthesis [84], it is possible that heroin-induced adaptations in TSP-1 expression due to increased oxidative stress, and normalization of this by NAC, may be linked to the altered dendritic spine morphometric features observed herein. Overall, it is likely that NAC’s ability to normalize heroin-induced adaptations in astrocyte–synapse proximity and GFAP cytoskeletal architecture has a direct influence on structural plasticity in PrL-NAcore neurons, perhaps through indirect or direct mechanisms at neurons. Thus, the degree to which PAPs associate with dendritic spines of PrL-NAcore neurons, whether this is altered following the extinction of heroin SA, the involvement of TSP-1 versus TSP-2, and whether NAC has an impact are areas of future investigation.

NAC has previously been demonstrated to regulate glutamate homeostasis, which has been consistently shown to be disrupted following drug self-administration [15,16,17,24]. Indeed, the normalization of glutamate transmission and related adaptations in astrocyte structural plasticity have been shown to be centrally important in preventing relapse to drugs of abuse. As an example, treatment with other compounds that restore glutamate homeostasis and astrocyte interaction with synapses such as ceftriaxone have shown the capacity to block cued-relapse to cocaine [10]. Overall, it is likely that NAC’s ability to normalize heroin-induced adaptations in astrocyte–synapse proximity and GFAP cytoskeletal architecture have a direct influence on plasticity in PrL-NAcore neurons, perhaps through indirect or direct mechanisms at neurons mediated by extracellular glutamate concentrations and/or the levels of physical interaction of astrocytes with neurons. Accordingly, the degree to which heroin impacts PAPs’ association with dendritic spines of PrL-NAcore neurons, whether this is altered following the extinction of heroin SA, the involvement of astrocyte signaling mechanisms, and whether NAC treatment reverses drug-induced adaptations in this pathway remain future directions for investigation.

### 4.7. Limitations and Methodological Considerations

The most obvious limitation to the present set of experiments is the omission of female rodents. These series of studies were initially planned as a follow-up to existing literature surrounding astrocyte–neuron interactions in the NAcore [12]. Because these initial experiments only used male animals, we omitted females from the present set of experiments as these earlier findings guided a priori power investigations and study planning. We fully recognize the importance and implications of sex differences in addiction, and plan to address this shortcoming by evaluating female rodents in identical experimental parameters.

Another limitation of the present findings is that they are largely correlative in nature, and our present experimental manipulations do not allow us to infer causality between heroin adaptations to neurons and astrocytes, cued-relapse to heroin, and the modulatory role that NAC plays in this process. Future experiments will involve targeted manipulations of these specific features in an attempt to establish causal relationships between the relationships discussed.

While we show that the evoked astrocytic GLT-1-dependent current can be a predictive measurement of prior heroin exposure, it is unclear as to the functional ramifications this effect has owing to the relatively marginal differences observed between the evoked GLT-1-dependent current between YS- and heroin-treated animals. Moreover, as discussed above, it is unclear if any such alterations play a role in cued-relapse to heroin.

Another limitation of the present experiments was the lack of YS control animals treated with NAC. In the current study, we chose to omit YS animals injected with NAC for several reasons. Most importantly, NAC has been shown to have no effect on NAcore glutamate concentrations or cystine uptake through Xc- in control animals [48]. Moreover, NAC treatment normalized increased glutamate concentrations in the anterior cingulate of cocaine-addicted patients but had no effect in healthy controls [85].

Canonically, GFAP is thought to only be present in the major branches and endfeet of astrocytes and not to be present in all non-reactive astrocytes in adult healthy CNS tissue [86]. However, our data are potentially at odds with this view as we found consistent fields of astrocytes in all three experimental groups and did not observe gaps that could potentially represent GFAP-negative astrocytes. As described above, GFAP does not extend into the finest membranous processes, which are the primary domains of astrocyte–neuron interactions, the motility of which are regulated by actin cycling [87]. The ability to use membrane-targeted expression of EGFP driven by astrocyte-specific promoters in conjunction with the immunohistochemical detection of synaptic proteins has gained traction as an assay to determine the proximity of PAPs to synapses (see [53,54] for reviews). Moreover, GFAP comprises ~15% of total astrocyte volume, which has historically led to incorrect conclusions regarding the domains of individual astrocytes, which further suggests the necessity of this approach in the accurate estimation of the astrocyte–neuron contact [61]. In this study, we used two different synaptic markers, Synapsin I and GluA2, which label generic pre-synaptic terminals (Synapsin I) and excitatory post-synaptic AMPA receptors (GluA2), respectively. However, it is important to note that, with an approximate minimum resolution of 150 nm, we are not attempting to distinguish between pre- and post-synaptic elements, separated by an approximate distance of 10 nm [88]. We are merely using the IHC detection of these proteins as a means to label synapses. However, the fact that both antibodies yielded the increased co-registration of synapses with astrocytic membranes in heroin animals can be viewed as a means to exclude epiphenomenological or potentially off-target effects from each individual antibody.

Finally, in our shTSP2 experiments, we utilized the construct provided by the Dong Lab. Validation of TSP2 knockdown was performed by the Dong Lab in HEK293 cells and reduced cocaine-mediated synaptogenesis in vivo. [30]. Quantitative analysis of the magnitude of TSP2 knockdown would require alternative techniques such as RNAScope for TSP2 and an astrocyte marker, which is a limitation of our study. In addition, our experiment utilized the AAV5. *GfaABC1D*::Lck-EGFP virus as a negative control as opposed to an off-target shRNA construct which is also a limitation of the present study.

## 5. Conclusions

In conclusion, our data indicate that heroin SA and extinction elicit hypertrophic complexity and synapse association of astrocytes in the PrL cortex that are associated with altered dendritic spine morphological features in PrL-NAcore neurons. Our data fill a much-needed gap in the field by focusing on the upstream portion of the canonical relapse circuit to accompany what has been observed in the NAcore. Moreover, these findings point to an additional node of this circuitry whereby NAC can modify heroin-induced adaptations in the PrL cortex in addition to its well-defined modification of NAcore synaptic pathology. Ultimately, astrocytic and neuronal adaptations in the PrL cortex may be a target for informing future drug discovery for relapse prevention pharmacotherapies.

## Figures and Tables

**Figure 1 cells-12-01812-f001:**
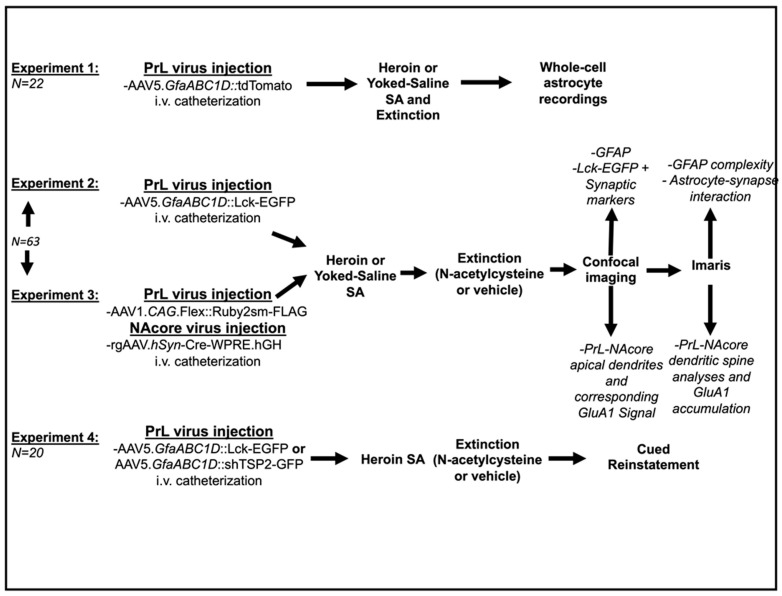
Diagram illustrating the experimental timeline for each experiment.

**Figure 2 cells-12-01812-f002:**
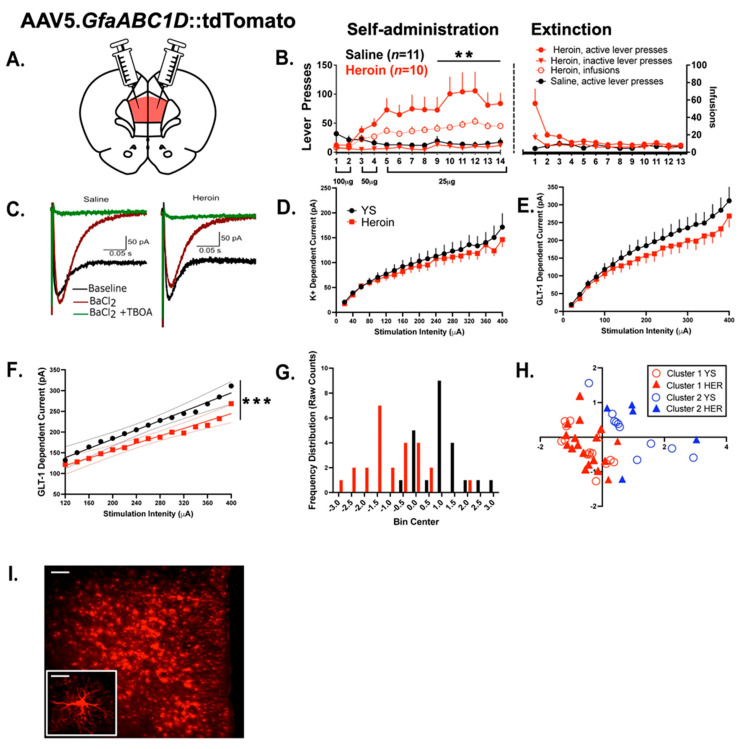
Evoked GLT−1-dependent current following heroin SA and extinction reveals functional clustering of PrL astrocyte populations. (**A**) Schematic indicating viral vector design. (**B**) Self-administration and extinction behavior for heroin and YS rats. Days of SA and extinction are displayed on the x-axis. Lever presses are shown on the left y-axis and infusions are shown on the right y-axis. Note that the infusion scale in panel B refers only to the self-administration data; no cue presentations of infusions occur during the extinction process. The statistical difference between active lever presses for rats receiving heroin and those receiving saline is indicated. (**C**) Representative traces of GLT−1 and K^+^ current in PrL astrocytes following a single stimulation in saline- or heroin-treated rats. (**D**) Evoked K^+^ current. (**E**) Evoked GLT−1 dependent current. (**F**) Isolation of area of interest and nonlinear modeling of YS (black)- vs. heroin (red)-receiving animals. (**G**) Discriminant scores for YS (black)- versus heroin (red)-receiving animals. (**H**) Functional clustering of PrL astrocyte populations. (**I**) Representative low magnification image of AAV5.*GfaABC1D*::tdTomato expression in the prelimbic cortex with high magnification inset. Low magnification scale bar = 100 µm, high magnification scale bar = 5 µm. ** *p* < 0.01, *** *p* < 0.001.

**Figure 3 cells-12-01812-f003:**
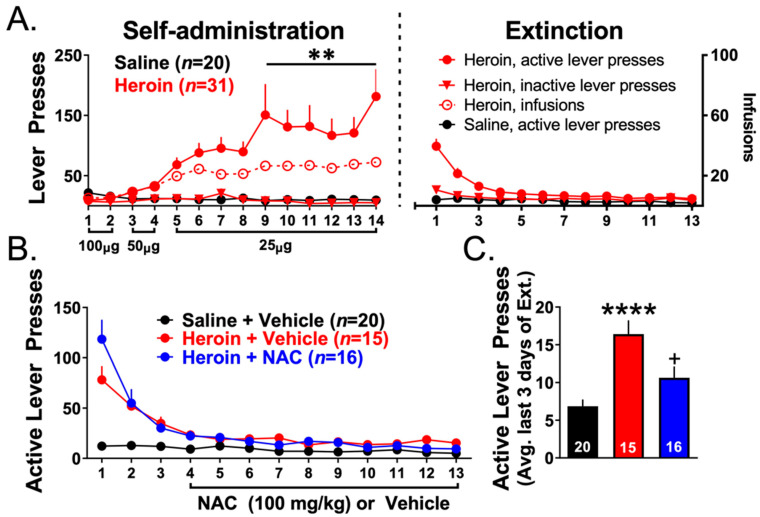
NAC reduces extinction responding. (**A**) Self-administration and extinction behavior for heroin and YS rats. Lever presses are shown on the left y-axis and infusions are shown on the right y-axis. ** *p* < 0.01 compared to saline active lever presses. (**B**) Active lever presses during extinction for YS, heroin animals injected with vehicle, and heroin animals injected with NAC. (**C**) Average active lever presses over the last 3 days of extinction for the three experimental groups. **** *p* < 0.0001 compared to YS, + *p* < 0.05 compared to Heroin + Vehicle. The number of animals within each group are shown within the bars.

**Figure 4 cells-12-01812-f004:**
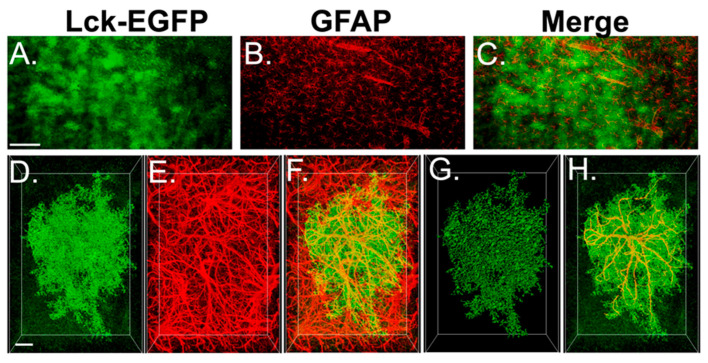
AAV5.*GfaABC1D*::Lck-EGFP expression is specific to GFAP+ astrocytes. (**A**–**C**) Low magnification view of AAV5.*GfaABC1D*::Lck-EGFP (green), GFAP (red), and the merge of the two signals. (**D**–**F**) Raw deconvolved GFP, GFAP immunohistochemical detection, and the merge of the two signals. (**G**) 3D space-filling model (green) to isolate the EGFP+ astrocyte from background. (**H**) GFAP signal (red) confined to the isolated GFP+ astrocyte (green). Scale bars indicate either 150 µm (**A**–**C**) or 10 µm (**D**–**H**).

**Figure 5 cells-12-01812-f005:**
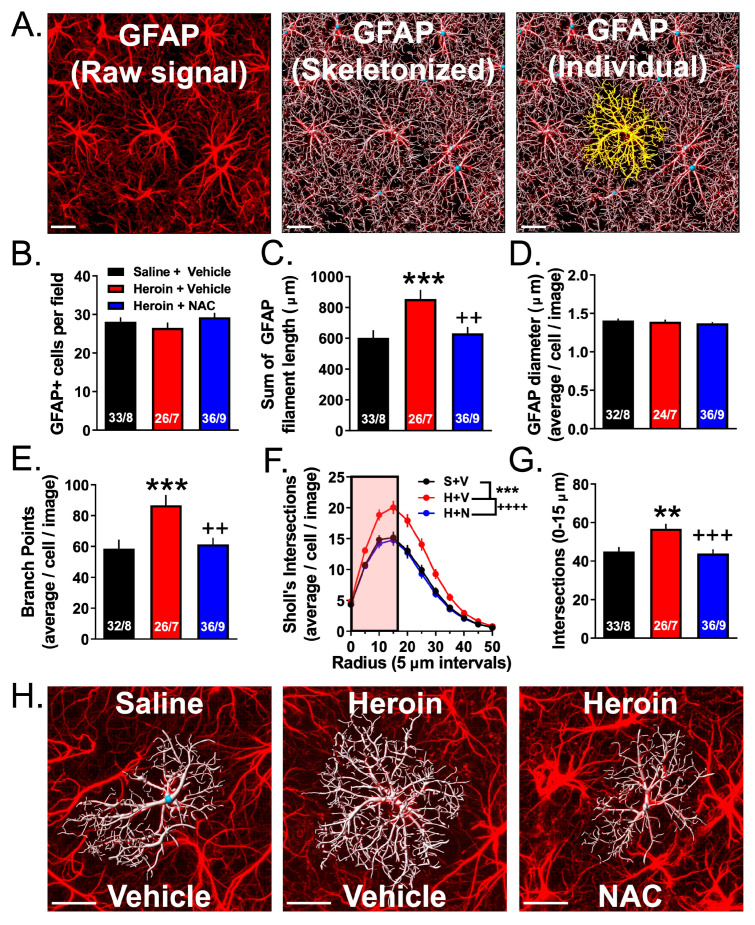
NAC normalizes heroin-induced alterations in PrL cortical astrocyte complexity. (**A**) Representative field of GFAP+ astrocytes skeletonized for analysis. Left—Raw GFAP signal from the confocal microscope. Middle—Skeletonized GFAP. Right—Isolated GFAP+ astrocyte from the field of cells. (**B**) There was no difference between groups in the number of GFAP+ cells per field. (**C**) Heroin increased the total length of the GFAP cytoskeletal filament which was normalized by repeated NAC treatment. (**D**) There was no difference between groups in the average diameter of the GFAP filament. (**E**) Heroin increased the average number of branch points which was normalized by repeated NAC treatment. (**F**) Frequency distribution of the number of Sholl intersections as a function of distance from the center of the cell. S + V = YS + Vehicle, H + V = Heroin + Vehicle, H + N = Heroin + NAC (**G**) Average number of Sholl intersections across the first 4 bins (0, 5, 10, and 15 µm). (**H**) Representative skeletonized GFAP+ cells for each treatment group. ** *p* < 0.01, *** *p* < 0.001 compared to YS, ++ *p* < 0.01, +++ *p* < 0.001, ++++ *p* < 0.0001 compared to Heroin + Vehicle. Numbers within each bar (**B**–**D**) denote the number of images taken/the number of animals used. Numbers within each bar (**E**,**G**) indicate the number of individual astrocytes processed/total number of animals in each condition. Scale bars indicate 20 µm.

**Figure 6 cells-12-01812-f006:**
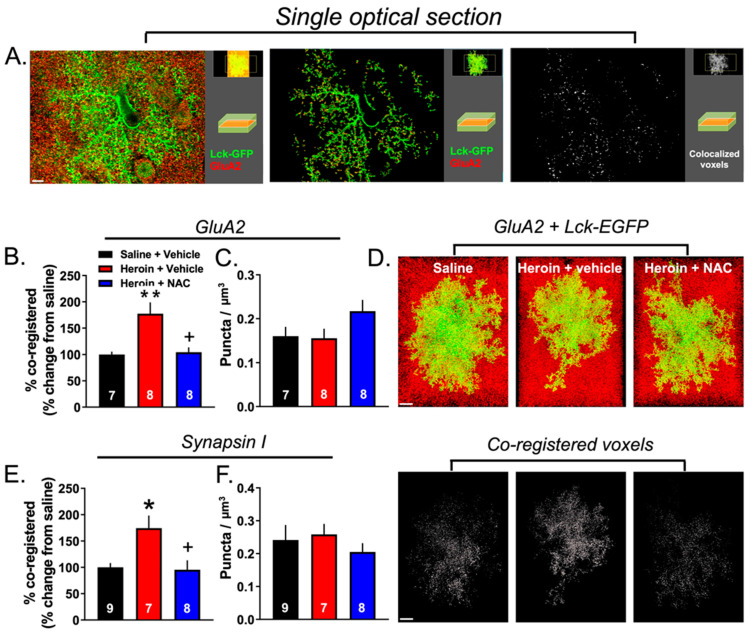
NAC normalized heroin-induced increases in PrL cortical astrocyte PAP–synapse association. (**A**) A single optical section from a representative Lck-EGFP-labeled astrocyte and GluA2 puncta. Left—Raw signal from confocal microscope. Middle—GluA2 confined within the proximity of the astrocyte membrane. Right—Co-registered voxels within the proximity of the astrocyte membrane. (**B**) Percent of co-registration of GFP and GluA2. (**C**) Number of GluA2 puncta normalized to the data-set volume. (**D**) Representative astrocytes from the three groups. Top—Raw signal from the confocal microscope. Bottom—Co-registered voxels within the region occupied by each astrocyte. (**E**) Heroin SA and extinction increased the percent co-registration of GFP and Synapsin I which was normalized by repeated NAC treatment (**F**) There was no difference between groups in the number of Synapsin I puncta normalized to the data-set volume. Numbers within each bar denote the number of animals used, black bars indicate Saline + Vehicle treatment, red bars indicate Heroin + Vehicle Treatment, blue bars indicate Heroin + NAC treatment. * *p* < 0.05 ** *p* < 0.01 compared to YS, + *p* < 0.05 compared to Heroin + Vehicle. Scale bars indicate 5 µm (**A**) and 10 µm (**D**).

**Figure 7 cells-12-01812-f007:**
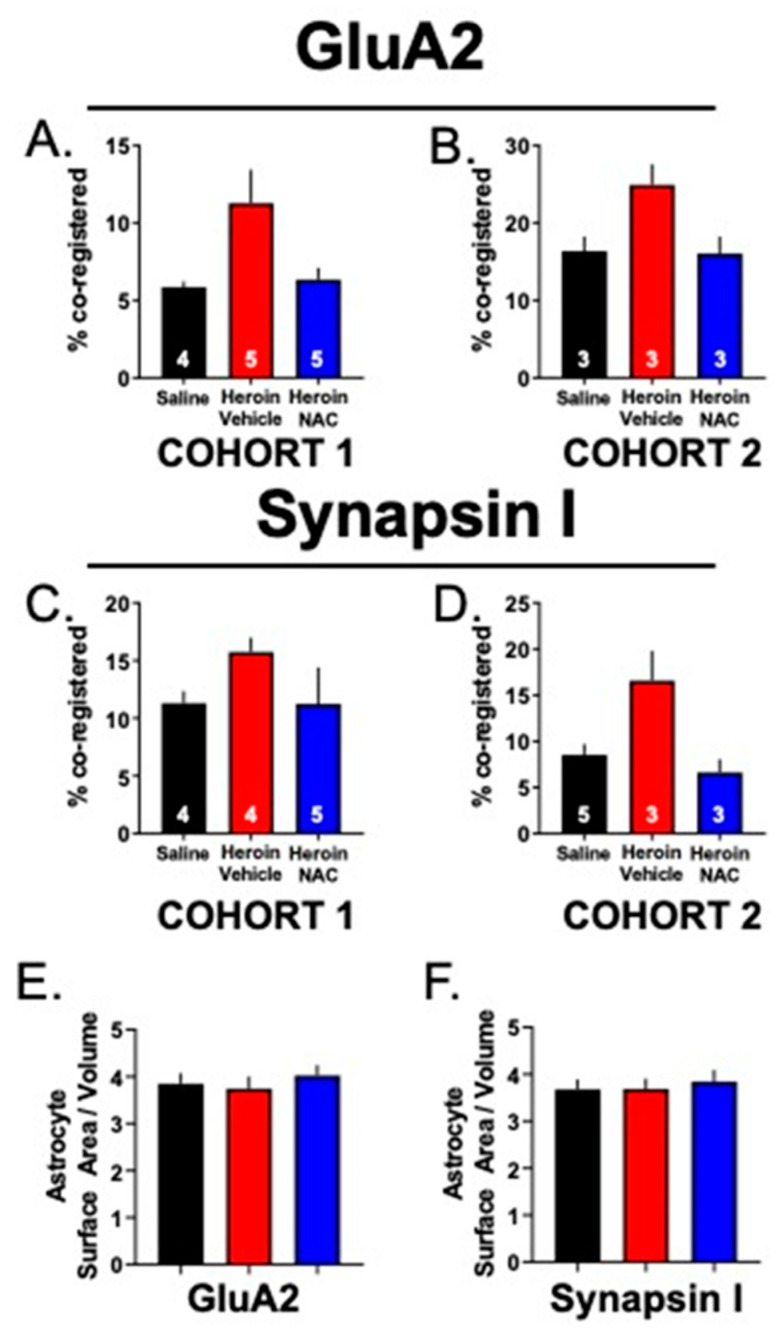
Between-cohort effects on GluA2 and Synapsin I co-registry with AAV5.*GfaABC1D*::LcK-EGFP. (**A**,**B**) Cohort 1 (**A**) and cohort 2 (**B**) showed the same effect of percent co-registry for GluA2. (**C**,**D**) Cohort 1 (**C**) and cohort 2 (**D**) showed the same effect of percent co-registry for Synapsin I. (**E**,**F**) 3D-rendered AAV5.*GfaABC1D*::LcK-EGFP-expressing astrocytes showed no difference between groups in the surface-area-to-volume ratio for GluA2 (**E**) or Synapsin I (**F**). Black bars indicate Saline + Vehicle treatment, red bars indicate Heroin + Vehicle Treatment, blue bars indicate Heroin + NAC treatment.

**Figure 8 cells-12-01812-f008:**
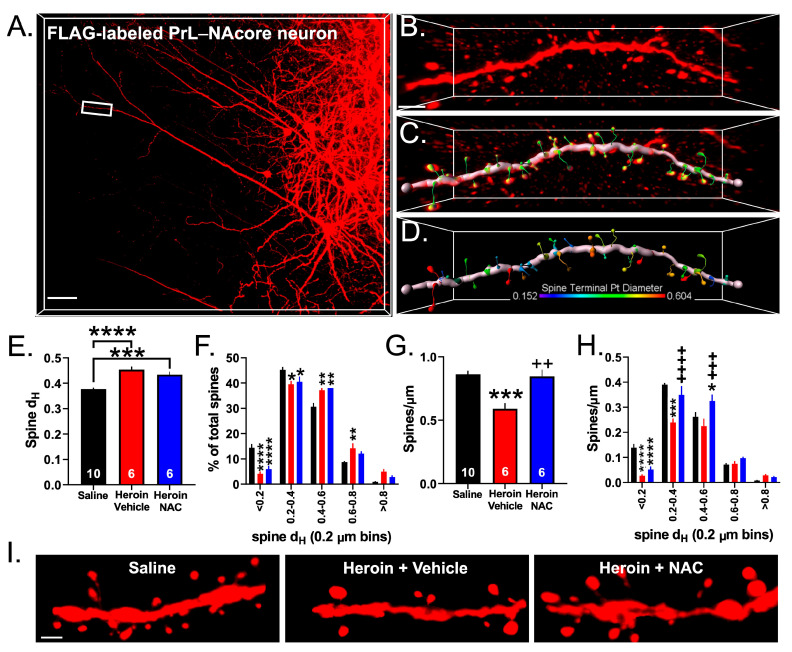
NAC normalized heroin-induced decreases in PrL–NAcore apical spine density, but not head diameter. (**A**) Left—Representative FLAG-labeled PrL–NAcore neuron. (**B**) Dendritic spine segment taken from the apical tuft (boxed region in **A**), (**C**) Filament-processed spine segment, (**D**) Heat map of dendritic spine head diameters along the segment—warmer colors indicate a greater head diameter. Note that most spines fall within the middle range of head diameters. (**E**) Average d_H_ across treatment groups. (**F**) Frequency distribution of the percent of total spines as a function of spine head diameter bin (0.2 µm). (**G**) Average dendritic spine density across treatment groups. (**H**) Frequency distribution of spine density normalized to spine d_H_. (**I**) Representative spines from a 15 µm stretch of dendrite are shown. d_H_ = head diameter. Numbers within each bar denote the number of animals used. * *p* < 0.05, ** *p* < 0.01, *** *p* < 0.001, **** *p* < 0.0001 compared to YS. ++ *p* < 0.01, +++ *p* < 0.001, ++++ *p <* 0.0001 compared to Heroin + Vehicle. Scale bars indicate 70 µm (**A**), 3 µm (**B**–**D**), and 2 µm (**I**).

**Figure 9 cells-12-01812-f009:**
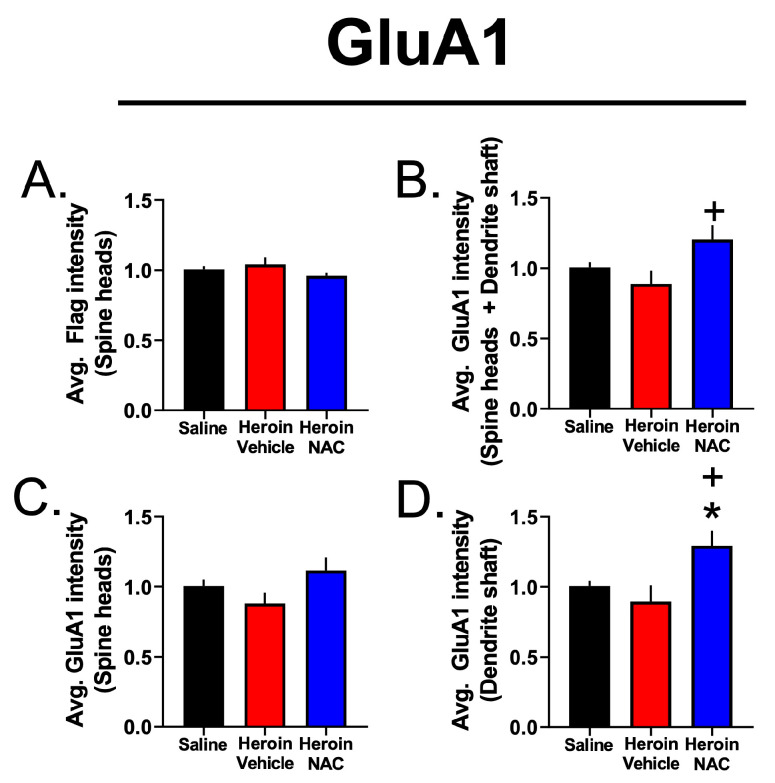
Signal intensity data pertaining to Experiment 3. (**A**) Average FLAG intensity in dendritic spine heads across treatment groups. (**B**) Average GluA1 intensity (dendritic shaft and spine head) across treatment groups. (**C**) Average GluA1 intensity isolated to dendritic spine heads. (**D**) Average GluA1 intensity isolated to dendritic shafts. ** p <* 0.05 compared to YS, *+ p <* 0.05 compared to Heroin + Vehicle. Data are expressed as a fold change from control.

**Figure 10 cells-12-01812-f010:**
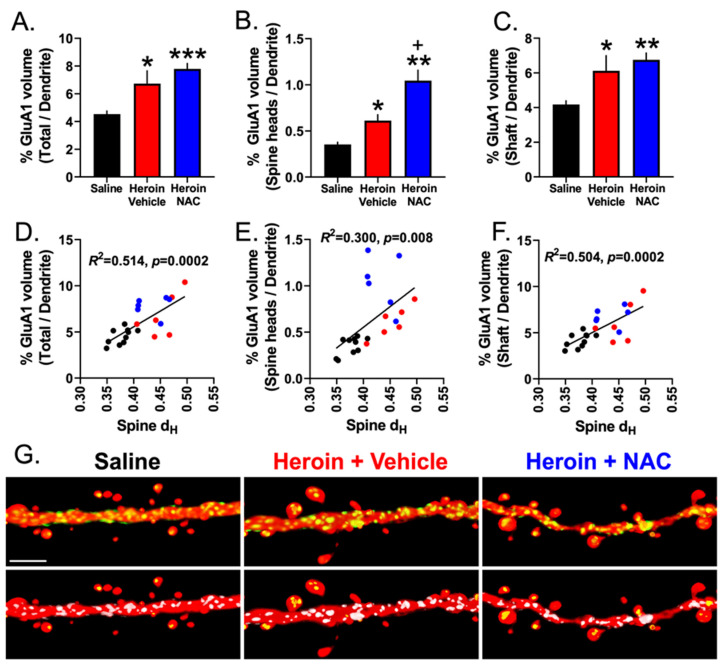
Heroin SA and extinction, regardless of NAC treatment, increased GluA1 immunoreactivity in dendritic shafts and spine heads of PrL-NAcore neurons. (**A**) Average GluA1 immunoreactivity in the total dendrite (shaft + spine head) normalized to dendrite volume. (**B**) Average GluA1 immunoreactivity in dendritic spine heads normalized to dendrite volume. (**C**) Average GluA1 immunoreactivity in the dendritic shaft normalized to dendrite volume. (**D**) Scatterplot depicting correlation between spine d_H_ and GluA1 immunoreactivity in the total dendrite (shaft + spine head) normalized to dendrite volume. (**E**) Scatterplot depicting correlation between spine d_H_ and GluA1 immunoreactivity in dendritic spine heads normalized to dendrite volume. (**F**) Scatterplot depicting correlation between spine d_H_ and GluA1 immunoreactivity in the dendritic shaft normalized to dendrite volume. Black lines indicate linear fit obtained from linear regression analyses. (**G**) Top panel—Representative dendritic spine segments showing Flag (red) and GluA1 (green). For clarity, GluA1 is only shown in the volume occupied by the dendrite segment. Bottom panel—Co-registry between GluA1 and Flag in the dendritic shaft (white) and spine heads (yellow). * *p* < 0.05, ** *p* < 0.01, *** *p* < 0.001 compared to Saline; + *p* < 0.05 compared to Heroin + Vehicle. Scale bar indicates 4 µm. (**D**–**F**) black points indicate Saline + Vehicle treatment, red points indicate Heroin + Vehicle Treatment, blue points indicate Heroin + NAC treatment.

**Figure 11 cells-12-01812-f011:**
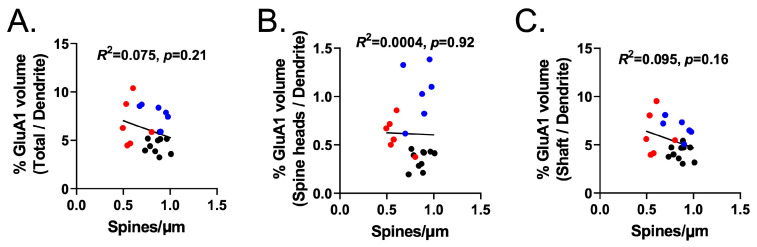
There was no correlation between the percent GluA1 volume in subcompartments normalized to the total dendrite volume and spine density. (**A**) % GluA1 volume in the co-registered volume within the total dendrite: y = −3.488X + 8.757, r = −0.27 r^2^ = 0.075. (**B**) % GluA1 volume in the co-registered volume within the spine heads only: y = −0.043X − 0.647, r = −0.02 r^2^ = 0.0004. (**C**) % GluA1 volume in the co-registered volume within the dendritic shafts only: y = −3.445X + 8.110, r = −0.31 r^2^ = 0.095. Black lines indicate linear fit. Black points indicate Saline + Vehicle treatment, red points indicate Heroin + Vehicle Treatment, blue points indicate Heroin + NAC treatment.

**Figure 12 cells-12-01812-f012:**
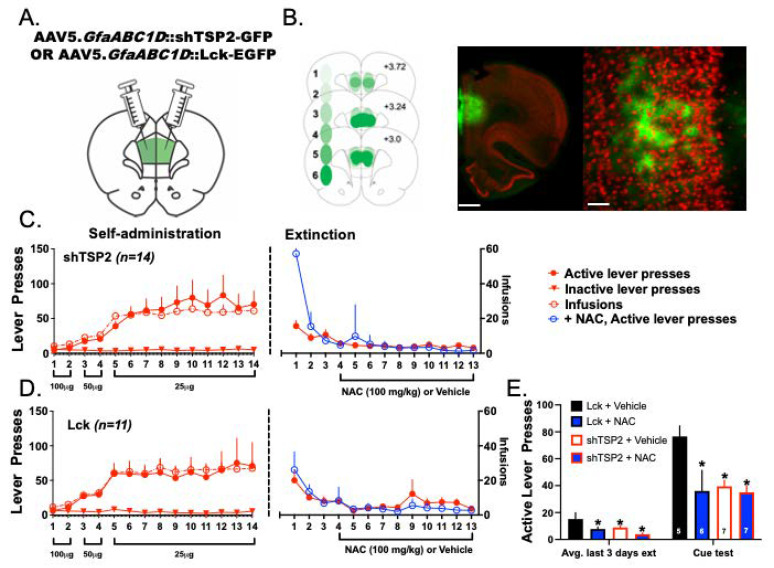
NAC treatment or loss of TSP2 on cortical astrocytes blunts cue-induced relapse to heroin. (**A**) Schematic indicating viral design. (**B**) Schematic indicating viral placements with approximate anterior/posterior viral spread from the injection site and representative low- and high-magnification images showing GFP expression in the PrL and viral labeling of astrocytes (green) versus neuN counterstain (red) (scale bars: low mag = 1500 µm, high mag = 100 µm). (**C**) Self-administration and extinction data for animals injected with shTSP-2. (**D**) Self-administration and extinction data for animals injected with Lck-EGFP. (**E**) Effects for the last three days of extinction and for cued-relapse to heroin seeking following extinction. ** p <* 0.05 compared to Lck + Vehicle.

## Data Availability

Not applicable.

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
