# Peer review of "Heroin Self-Administration and Extinction Increase Prelimbic Cortical Astrocyte–Synapse Proximity and Alter Dendritic Spine Morphometrics That Are Reversed by N-Acetylcysteine"

_cells, 2023, doi:10.3390/cells12141812_

Round 1
Reviewer 1 Report
This is a well-written and provocative study demonstrating the impact of heroin on PrL-NAc circuitry and spine dynamics. The model is novel and its effects on cortical astrocyte complexity and extinction were largely consistent. I found the experimental protocols in general technically sound, and the statistical analysis and experimental power to be appropriate. My primary concerns are geared around the usage of NAC as a therapeutic intervention.
1. NAC being an antioxidant, how it reversed astrocytic complexity, dendritic spine morphology, and GluA1 expression? Although arguably the authors mentioned it might be due to oxidative stress, there are no supporting data for that claim in this paper.
2. Minor: There are several spell checks to be performed.
Author Response
Reviewer 1:
This is a well-written and provocative study demonstrating the impact of heroin on PrL-NAc
circuitry and spine dynamics. The model is novel and its effects on cortical astrocyte complexity and extinction were largely consistent. I found the experimental protocols in general technically sound, and the statistical analysis and experimental power to be appropriate. My primary concerns are geared around the usage of NAC as a therapeutic intervention.
- NAC being an antioxidant, how it reversed astrocytic complexity, dendritic spine morphology,
and GluA1 expression? Although arguably the authors mentioned it might be due to oxidative stress, there are no supporting data for that claim in this paper.
2. Minor: There are several spell checks to be performed
We thank the reviewer for their positive comments, and agree that more description of NAC’s actions at astrocytes would improve the overall clarity of the manuscript. NAC’s mechanisms of activity are multifaceted. In addition to its antioxidant activity, NAC serves as a substrate for the cystine-glutamate exchanger expressed in astroglia. Indeed, in the nucleus accumbens NAC has previously been shown drive cystine-glutamate exchange and to restore basal levels of glutamate in the nucleus accumbens core following cocaine withdrawal, an effect likely linked to NAC’s ability to also restore expression of the glutamate transporter and prevent drug seeking. Thus, we suspect that NAC’s ability to reverse heroin-mediated alterations in astrocytic complexity , dendritic spine morphology and GluA1 expression in the prelimbic cortex are linked to the ability of NAC to promote cortical astrocyte homeostatic function. We have expanded our discussion in the text to address this concern.
Reviewer 2 Report
The article presents data on efficacy of N-acethylcysteine (100mg/kg, daily) in prevention of heroine cue-induced seeking in rats. This particular data is not novel itself, as this effect was published previously as authors indicate (ref. 19). The novel results pertains the fact that NAC is effective in restoring the normal astrocyte arborisation and contact with synapses, as well as subtle dendritic spine morphology, which are dysfunctional in heroine treated rats. Authors discuss that NAC effect may result from normalisation of extracellular glutamate, what may be either primary or secondary to astrocyte remodelling.
The separate part, in my reception, is the thrombospondin experiment. I do not see a clear explanation, why TSP2 downregulation was implemented – e.g. how does it mechanistically correlate with NAC potential mechanism of action? Since we do not have astrocytic morphology after TSP2 silencing, I would consider deleting this part from the manuscript. Or, perhaps, heroine abuse could affect the TSP1/2 expression?
I do not have any comments concerning methodology, experiment design nor the manuscript language.
Author Response
The article presents data on efficacy of N-acetylcysteine (100mg/kg, daily) in prevention of heroine cue-induced seeking in rats. This particular data is not novel itself, as this effect was published previously as authors indicate (ref. 19). The novel results pertains the fact that NAC is effective in restoring the normal astrocyte arborization and contact with synapses, as well as subtle dendritic spine morphology, which are dysfunctional in heroine treated rats. Authors discuss that NAC effect may result from normalization of extracellular glutamate, what may be either primary or secondary to astrocyte remodeling.
The separate part, in my reception, is the thrombospondin experiment. I do not see a clear explanation, why TSP2 downregulation was implemented – e.g. how does it mechanistically correlate with NAC potential mechanism of action? Since we do not have astrocytic morphology after TSP2 silencing, I would consider deleting this part from the manuscript. Or, perhaps, heroine abuse could affect theTSP1/2 expression?
We apologize for the confusion with the rationale for the inclusion of the TSP experimentation and agree that this issue needs clarification within our manuscript. Given that we observed a heroin-induced reduction in dendritic spine density at PrL-NAcore neurons, and that NAC both reversed this effect, and prevents heroin seeking, we hypothesized that the therapeutic efficacy of NAC may require astrocyte-mediated synaptogenesis. As glial generated thrombospondin signaling is required for the formation of new synapses, we sought to evaluate if TSP2 expression in cortical astrocytes was required for NACs ability to inhibit cue-induced heroin seeking. While this was not what we observed experimentally, we have expanded upon this within the results and discussion in the manuscript to better provide the reader with the rational for the inclusion of these data. We feel that the data are however important to retain within the study as it demonstrates that manipulation of gene expression with cortical astroglia is sufficient to limit cue-induced heroin seeking.